evolution, genetics

mimicry, *WntA*, developmental constraints, convergence, butterflies

**Author for correspondence:**
Steven M. Van Belleghem
e-mail: vanbelleghemsteven@hotmail.com

# Perfect mimicry between *Heliconius* butterflies is constrained by genetics and development

Steven M. Van Belleghem[1], Paola A. Alicea Roman[1,2],
Heriberto Carbia Gutierrez[1,3], Brian A. Counterman[4] and Riccardo Papa[1,5,6]

[1]Department of Biology, University of Puerto Rico, Rio Piedras, Puerto Rico
[2]Department of Biology, University of Puerto Rico, Humacao, Puerto Rico
[3]Department of Mathematics, University of Puerto Rico, Rio Piedras, Puerto Rico
[4]Department of Biological Sciences, Mississippi State University, Mississippi State, MS, USA
[5]Smithsonian Tropical Research Institution, Panama, Republic of Panama
[6]Molecular Sciences and Research Center, University of Puerto Rico, Puerto Rico

SMVB, 0000-0001-9399-1007; PAAR, 0000-0002-2516-1893; HCG, 0000-0002-2415-646X;
BAC, 0000-0003-2724-071X; RP, 0000-0002-7986-9993

Müllerian mimicry strongly exemplifies the power of natural selection. However, the exact measure of such adaptive phenotypic convergence and the possible causes of its imperfection often remain unidentified. Here, we first quantify wing colour pattern differences in the forewing region of 14 co-mimetic colour pattern morphs of the butterfly species *Heliconius erato* and *Heliconius melpomene* and measure the extent to which mimicking colour pattern morphs are not perfectly identical. Next, using gene-editing CRISPR/Cas9 KO experiments of the gene *WntA*, which has been mapped to colour pattern diversity in these butterflies, we explore the exact areas of the wings in which *WntA* affects colour pattern formation differently in *H. erato* and *H. melpomene*. We find that, while the relative size of the forewing pattern is generally nearly identical between co-mimics, the CRISPR/Cas9 KO results highlight divergent boundaries in the wing that prevent the co-mimics from achieving perfect mimicry. We suggest that this mismatch may be explained by divergence in the gene regulatory network that defines wing colour patterning in both species, thus constraining morphological evolution even between closely related species.

## 1. Introduction

Adaptation is the product of natural selection on the genetic and phenotypic diversity within a population [1]. In this regard, key developmental steps that limit or bias trait variation can pose so-called constraints on the directionality of evolution [2–4]. Consequently, when populations evolve independently, lineages can accumulate changes that lead evolution along an irreversible trajectory [5]. Understanding the relative contribution (or constraint) of genetics and development to adaptation would therefore allow us to better comprehend the directionality and predictability of evolution [6].

To date, the extent to which evolutionary changes are consequential for future adaptation has been most elegantly studied using artificial selection experiments [1]. In these experiments, multiple generations can be relatively easily traced while being exposed to contrasting selection pressures. The consequences of their adaptations can then be investigated when these selection pressures are reversed. In malaria, for example, a single mutation of large effect in the K13 protein can confer drug resistance [7] but simultaneously also favours the evolution of additional epistatic mutations [8]. As a consequence of these changes, the evolved phenotype cannot be simply reversed to its ancestral state when withdrawing

anti-malarial medication and thus has important consequences for resistance management strategies [8]. In butterflies, artificial selection on eyespots has been used to test for the existence of developmental constraints in wing colour patterns. Because these wing pattern elements belong to the same homologous series and share developmental pathways, covariances between them are expected in their size and shape. Nevertheless, in the butterfly *Bicyclus anynana*, artificial selection has suggested that there is great potential for independent change in size and shape of different eyespots [9]. This observation has been used to argue that natural selection plays a dominant role over developmental constraints in the observed eyespot diversity among *Bicyclus* species [9]. Similarly, in *Drosophila*, artificial selection experiments have shown that changes to wing shape can be rapidly induced from standing genetic variation present in the populations [10]. However, these induced phenotypes are lost when selection is suspended, presumably due to pleiotropic links to other traits that result in negative fitness consequences [11].

Not all study systems are amenable to perform artificial selection experiments. Alternatives to such controlled experiments are cases of convergent evolution that provide natural opportunities to investigate the selective, genetic, and developmental routes to adaptation [1]. For example, mimicry and the resulting phenotypic convergent evolution between distinct butterfly species provides a comparative framework to investigate the genes underlying the evolution and diversity of a wing colour pattern [12–14]. Recent studies have shown, for example, that the genes *WntA*, *cortex*, and *optix* are repeatedly used to control variation in wing colour patterns across Nymphalid butterflies [14–17]. Nevertheless, even in cases of Müllerian mimicry between species within the *Heliconius* genus, in which both partners have used the same genes to converge on an aposematic warning signal, some degree of imperfection in resemblance may exist. However, the precise extent to which *Heliconius* mimetic butterflies need to perfectly resemble the same phenotype to maximize the fitness value is not well understood. What may underlie these differences in resemblance are (i) conflicting or relaxed selection pressures and/or (ii) genetic and developmental constraints. Conflicting selective pressures can include variation in the mimicry community [18] and conflict between the outcomes of natural and sexual selection [19]. Relaxed selection pressures may result from coarse discrimination by predators [20,21]. On the other hand, genetic and developmental constraints can result from divergence in the genetic background or in the assembled gene regulatory network that affects the functioning and phenotypic effect of these genes.

*Heliconius erato* and *Heliconius melpomene* provide a textbook case of Müllerian mimicry. Although there is no evidence for gene flow between *H. erato* and *H. melpomene*, which split around 12–14 Mya [22], their resemblance in wing colour patterns is remarkable. This phenotypic convergence has evolved through strong selection pressures that benefit a common warning pattern that birds have learned to associate with unpalatability [23,24]. The discriminatory visual properties of birds appear to be quite precise, resulting in strong selection pressures for fine scale adjustments of the shape and size of colour patterns among local mimetic butterfly communities (for an overview of selection coefficients see [25]) [26,27].

Recently, a series of functional experiments have tested the role of the *WntA* gene in different *Heliconius* butterfly species and populations [12]. *WntA* is a member of the Wnt family of signalling ligands and a key molecular tool for butterfly wing colour pattern development. While the gene coding sequence is highly conserved across Lepidoptera, its *cis*-regulatory diversity underlies wing pattern shape variation between and within butterflies species [28,29]. Recent CRISPR/Cas9 KO experiments have shown its role in defining the ultimate colour fate of individual wing scale cells and highlighted incredible variability in the position and wing territory affected by *WntA* in *H. erato* and *H. melpomene* [12]. This result provides a visual representation of the effect of a divergent genetic background on a gene's function. In *H. erato* and *H. melpomene*, CRISPR/Cas9 *WntA* mutant phenotypes evidenced a more restricted area of modified black scales in the forewing of *H. melpomene* compared to *H. erato* [12,17]. This result demonstrates that although the two co-mimetic butterflies display identical forewing colourations, the genetic architecture and gene regulatory networks underlying their resemblance may be more complex than previously thought. These differences likely arise from divergence in the regulation of *WntA* and/or divergence in epistatic interactions with other genes, together defined as the gene regulatory network [12].

In the current study, we aimed to explore the contribution of genetic and developmental constraints to the phenotypic mismatch between co-mimetic pairs of *Heliconius* butterflies. We therefore first quantitatively measured and compared differences in the mid-forewing band (MFB) pattern of 14 co-mimicking populations of the butterfly species *H. erato* and *H. melpomene* from Central and South America. We then interpreted these differences in light of the recent *Heliconius WntA* CRISPR/Cas9 KO mutants [12]. We were specifically interested to test how species-specific divergence might impact the developmental function of *WntA* and limit adaptive convergence. More precisely, we argue that overlap of wild-type differences with differences in pattern boundaries as defined by *WntA* KOs in *H. erato* and *H. melpomene* would suggest the possible existence of genetic and developmental constraints for natural selection to achieve perfect mimicry. By combining the most recent methodological advances in functional experiments with quantitative measures, our study offers an alternative approach to artificial selection experiments to test the relative constraint of genetics and development to adaptation.

## 2. Materials

### (a) Colour pattern analysis

We obtained 8–14 images of each of 14 mimicking colour pattern morphs (electronic supplementary material, tables S1 and S2). Images were obtained through the authors' collections, collaborations, and collections made publicly available by Cuthill *et al.* [30] and Jiggins *et al.* [31]. Individual genders were determined based on sexual dimorphism in the androconial region [32]. To align images, we used a total of 11 landmarks at vein intersections on one forewing of each sample (electronic supplementary material, figure S1). Methods and results describing additional analysis to control for interspecific changes in wing shape and sex differences are described in Supplementary materials S1 and electronic supplementary material, table S3. While *WntA* is involved in black scale development in both the fore- and hindwing in *Heliconius* [12,17], differences in the distribution of *WntA* have mainly been found to correlate with the position of black

colour in the central part of the forewing among *Heliconius* colour pattern morphs, here called mid-forewing band or MFB [28]. With the interest of studying variation in the MFB, we extracted and focused on the area of the forewing in which black is absent and in which *WntA* is thus likely not expressed. MFB patterns were extracted and aligned using the R package *patternize* [33]. Depending on the MFB phenotype, we specified red, green, and blue (RGB) values for red, yellow, and/or white with a colour threshold (*colOffset*) chosen to fully extract the pattern. For *H. e. notabilis*, *H. m. plesseni*, and *H. m. cythera*, we extracted and combined both red and white to represent the MFB shape. Background noise or damaged regions in the wing that were co-extracted with the colour patterns were masked using the *setMask* function. Next, a thin plate spline (*tps*) transformation was obtained from transforming landmarks to a common reference sample. This common reference sample included the landmarks of an arbitrarily chosen sample and was used in all colour pattern analysis. The *tps* transformation was then used to align and compare the extracted MFB shape, size, and position.

Differences in the MFB patterns were first compared by subtracting the *H. erato* and *H. melpomene* pattern frequencies of each population, obtained with the *sumRaster* function in *patternize* (i.e. absolute MFB difference) and compared between co-mimics using a one-sample *t*-test. Next, the relative size of the pattern was calculated as the proportion of the total wing area in which the pattern is observed, using the *patArea* function (i.e. relative MFB difference) and compared between co-mimics using a two-sample *t*-test. Principal component analysis (PCA) was performed on the binary representation of the aligned colour pattern rasters of each sample [33]. The PCA visualizes the main variations in colour pattern boundaries among samples and groups and provides predictions of colour pattern changes along the PC axis, with positive values presenting a higher predicted presence of the MFB pattern and negative values presenting the absence of the pattern. Parts of the colour patterns that are present in all samples have a predicted value of zero, as these pixels do not contribute variance in the PCA. We tested the effect of population, sex, and species on shape variables using multivariate analysis of variance (MANOVA) and linear discriminant analysis (LDA) using only the values of samples along significant PC axes (see Supplementary Materials S1 for details).

## (b) *WntA* CRISPR KO analysis

Five mutant butterflies (10 wings) for each of the Panamanian geographical colour pattern morphs *Heliconius erato demophoon* and *Heliconius melpomene rosina* for which a frame shift mutation was generated at the gene *WntA* using CRISPR/Cas9 were obtained from Concha *et al.* [12]. All these mutants showed symmetric changes in wing patterns on both the left and right forewings and were thus likely full KO mutants [12]. Both left and right forewings were landmarked and the mutant pattern was extracted and aligned using the R package *patternize* [33]. Red was extracted from the *H. e. demophoon* mutants. As the *H. m. rosina* mutants often showed a yellow spot appearing in the proximal part of the MFB, both red and yellow were extracted for *H. m. rosina*. The mutant patterns were superimposed on the wild-type wing pattern comparisons by aligning to the common reference sample and using the *contour* function of the R package *raster* [34].

# 3. Results

## (a) Divergence and convergence in mid-forewing band pattern

Geographical colour pattern morphs of *H. erato* and *H. melpomene* cover a wide spectrum of MFB patterns, with unique or partially overlapping pattern elements among them (figure 1*a*,*b*). In the PCA of the MFB, the first main axis of variance was dominated by the absence or presence of a broad red MFB, also typically called a 'Postman' phenotype (PC1, 32% of variation; figure 1*c*). The second axis of variation in the MFB shape was dominated by the presence of either a narrow median band, as observed in *H. e. cyrbia*/*H. m. cythera*, *H. e. lativitta*/*H. m. malleti*, and *H. e. emma*/*H. m. aglaope*, or two spots, as observed in the *H. e. notabilis*/*H. m. plesseni* and *H. e. microclea*/*H. m. xenoclea* populations (PC2, 19% of variation; figure 1*c*).

As expected, co-mimicking populations of *H. erato* and *H. melpomene* are found to have more similar MFB phenotypes than different populations of the same species. However, significant differences in clustering can be observed in the PCA between the two species, with posterior probability of classification 88.6% and 92.2% for *H. erato* and *H. melpomene*, respectively ($F_{1,280} = 70.8$, $p < 0.001$; electronic supplementary material, table S4; figure 1*c*). When restricting our analysis to co-mimicking Postman populations, differences were also significant with posterior probability of classification 100% and 98.3% for *H. erato* and *H. melpomene*, respectively ($F_{1,127} = 70.0$, $p < 0.001$; electronic supplementary material, table S4; figure 1*d*). This difference between species was highly significant along PC2 (10% of variation; $F_{1,127} = 131.1$, $p < 0.001$) and suggests a general trend of expansion of both the proximal and distal area of the MFB in *H. melpomene* Postman phenotypes compared to *H. erato*. Significant differences between male and female MFB patterns were observed in both *H. erato* ($F_{1,139} = 3.11$, $p = 0.002$) and *H. melpomene* ($F_{1,140} = 3.17$, $p = 0.001$). However, sex differences had a low probability of posterior classification (55.8% and 58.3% for male and female *H. erato* and 78.0% and 62.5% for male and female *H. melpomene*, which is close to classification between random groups) and were only significant along PC axes that explain small amounts of variation among samples (electronic supplementary material, table S4).

Using pairwise comparisons we further explored interspecific differences between *H. erato* and *H. melpomene* co-mimics (figure 2*a*). First, among Postman phenotypes we observed an area in the distal posterior part of the MFB that consistently shows the absence of black scales in *H. melpomene* populations compared to *H. erato* populations (orange triangles in Postman (P) mimics in figure 2*a*). As demonstrated in the PCA on Postman phenotypes (figure 1*d*), we generally observed an expansion of the proximal area of the MFB in *H. melpomene* compared to *H. erato* in the pairwise comparisons (figure 2*a*). However, this trend was reversed between the Postman populations *H. e. hydara* and *H. m. melpomene* in French Guiana. In Colombia, the Postman phenotype showed an expansion of the MFB in the distal area in *H. e. venus* compared to *H. m. vulcanus* (figure 2*a*).

As seen for the Postman populations, all other co-mimicking *Heliconius* colour pattern morphs showed marked differences in their MFB (figure 2*a*). Comparison of the similarly patterned *H. e. cyrbia* with *H. m. cythera* from West Ecuador suggests that the position of the MFB is shifted proximally in *H. e. cyrbia* compared to *H. m. cythera*. In East Ecuador, the *H. m. malleti* generally

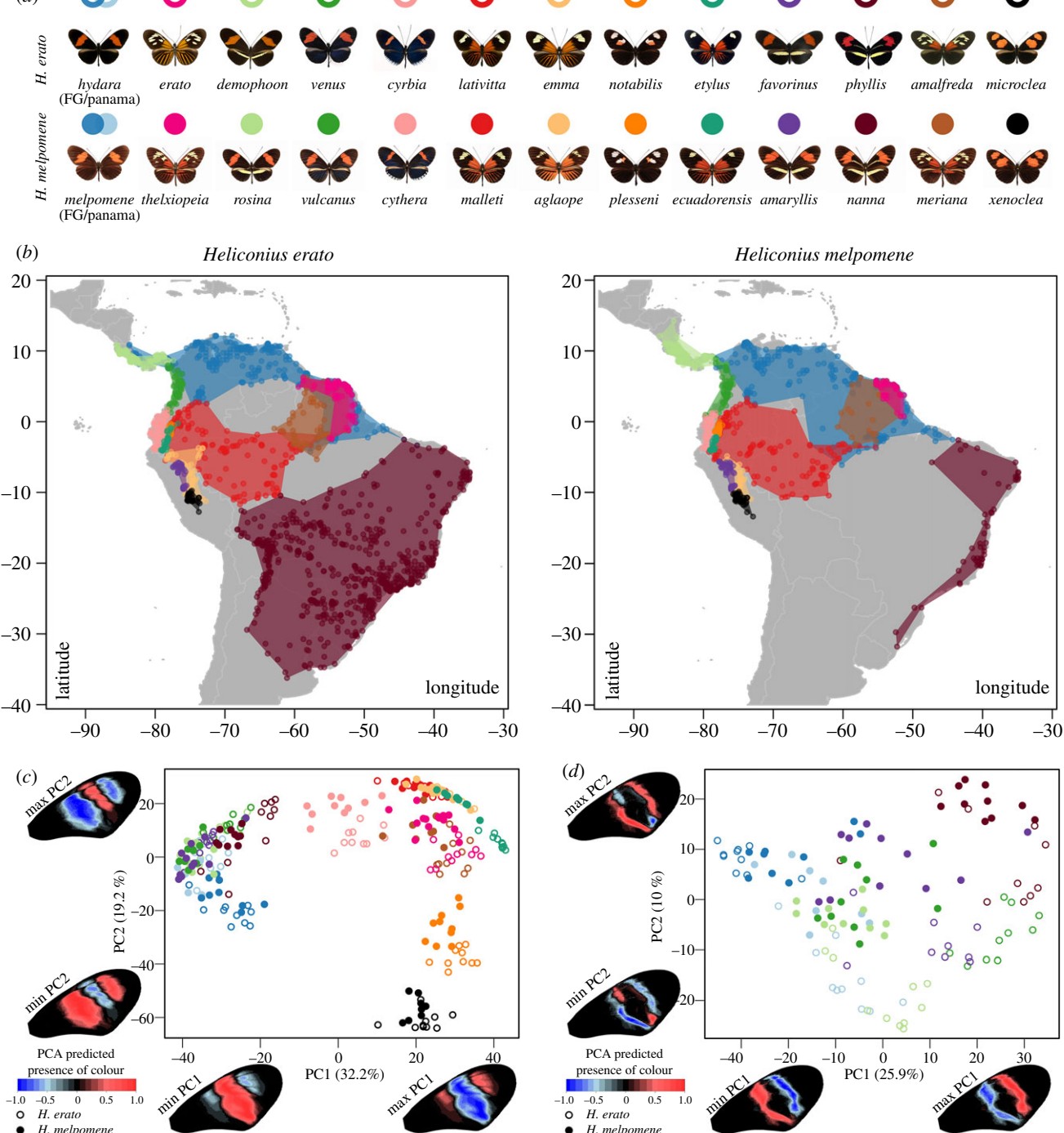

**Figure 1.** Co-mimicking *Heliconius erato* and *Heliconius melpomene* colour pattern morphs, their distribution and PCA of MFB pattern. (*a*) Dorsal images of mimicking *H. erato* and *H. melpomene* colour pattern morphs. (*b*) Distribution areas of the mimicking populations as obtained from Rosser *et al.* [35]. (*c*) PCA of mid-forewing band (MFB) shape of the mimicking colour pattern morphs. (*d*) PCA of MFB shape of mimicking 'Postman' populations. Wing heatmaps indicate minimum and maximum predicted MFB patterns along each PC axis while considering the PC value of all other PC axes at zero. Positive values present a higher predicted presence of the MFB pattern (red), whereas negative values present the absence of the pattern (blue). (Online version in colour.)

showed a larger MFB than the co-mimetic *H. e. lativitta* populations. Interestingly, differences in the comparison between the co-mimics *H. e. etylus* and *H. m. ecuadorensis*, which have a single distal spot, resembled differences in the distal spot between *H. e. notabilis* and *H. m. plesseni*. These latter populations have the so-called 'Split' MFB phenotype, which consists of two white/red spots in the MFB area ('S' in figure 2*a*). Finally, the so-called 'Broken' MFB phenotypes *H. e. erato* and *H. e. amalfreda*, which consist of multiple yellow spots in the MFB area ('B' in figure 2), consistently differed from the

*H. m. thelxiopeia* and *H. m. meriana* populations by a proximal shift of the distal margin.

In all co-mimetic comparisons of *H. erato* and *H. melpomene*, the average absolute difference which includes the position of the MFB pattern was larger than the average difference in the relative size of the MFB pattern (i.e. proportion of the wing in which MFB is present; figure 2*b*). Significant differences in the size of the MFB were only observed between the co-mimics *H. e. hydara* and *H. m. melpomene* from Panama ($p = 0.013$), *H. e. demophoon* and *H. m. rosina* ($p = 0.013$),

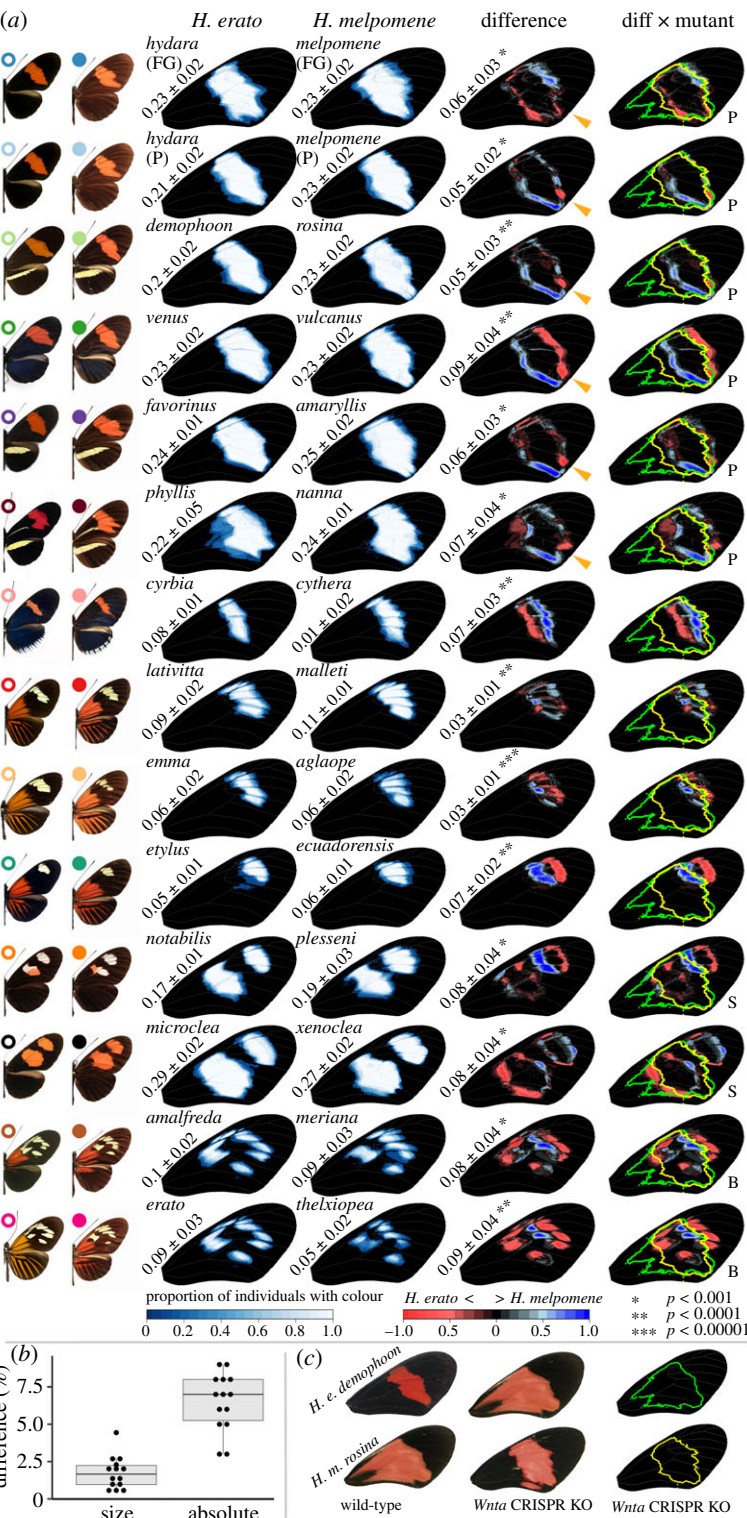

**Figure 2.** Quantification of mid-forewing band (MFB) pattern in co-mimicking *Heliconius erato* and *Heliconius melpomene* colour pattern morphs. (*a*) Heatmaps demonstrate the consistency of MFB within colour pattern morphs with white indicating consistent presence of MFB patterns and blue gradient indicating less consistent presence. Inter-species differences in MFB are shown on the right, with red indicating higher presence of MFB in *H. erato* and blue indicating higher presence of MFB in *H. melpomene*. Values next to wings indicate the average proportion of the wing in which the MFB is present within colour pattern morphs and in which differences are found between co-mimics. As a positional reference of the MFB pattern variation, the column on the right overlays the *H. e. demophoon* (green outline) and *H. m. rosina* (yellow outline) *WntA* CRISPR KO phenotype as found in at least 50% of the KO samples with the differences between co-mimics. 'P', 'B', and 'S' indicate phenotypes commonly referred to as the 'Postman', 'Broken', and 'Split' band morphs, respectively. Coloured circles next to butterfly wing images correspond to distribution areas in figure 1. (*b*) Comparison of the difference in relative size of the MFB (as proportion of the wing in which MFB is present) and absolute mismatch between *H. erato* and *H. melpomene* MFB. (*c*) Wild-type and *WntA* CRISPR KO phenotype of *H. e. demophoon* and *H. m. rosina*. (Online version in colour.)

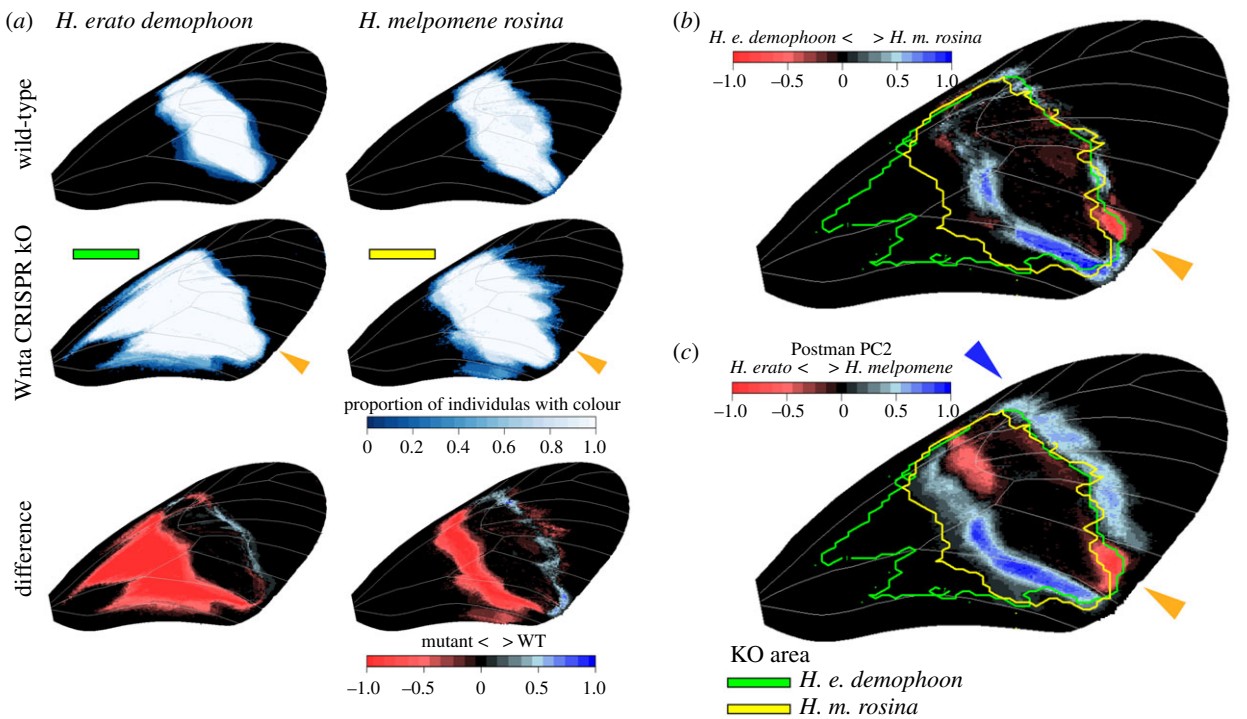

**Figure 3.** Comparison of mid-forewing band (MFB) differences between co-mimicking *Heliconius erato* and *Heliconius melpomene* Postman phenotypes and *WntA* CRISPR/Cas9 KO phenotypes. (*a*) Heatmaps showing the presence of MFB in wild-type, *WntA* KO phenotypes and their difference for *H. e. demophoon* and *H. m. rosina* (obtained from 10 wings of five KO butterflies each [12]). (*b*) Comparison of *WntA* KO area to wild-type difference between *H. e. demophoon* and *H. m. rosina*. (*c*) Comparison of *WntA* KO area to Postman phenotype variation (figure 1). Orange and blue arrows indicate overlap in mismatch between wild-type co-mimicking populations and *WntA* KO patterning area. The yellow and green outlines show the *WntA* KO area as present in at least 50% of the KO samples. (Online version in colour.)

*H. e. cyrbia* and *H. m. cythera* ( *p* = 0.009) and *H. e. erato* and *H. m. thelxiopeia* ( *p* = 0.001).

## (b) Mismatch between co-mimics partly coincides with developmental *WntA* boundaries

An important aspect of our study was to determine the possible link between wing pattern variation observed between wild-type butterflies and the *WntA* CRISPR/Cas9 KO phenotypes. We focused on the Panamanian co-mimics *H. e. demophoon* and *H. m. rosina* for which the largest *WntA* KO dataset is available and compared the *WntA* boundaries defined by these mutants with wild-type variation in the Postman phenotypes. As recently described by Concha *et al.* [12], *WntA* CRISPR/Cas9 KOs of the Postman phenotype *H. e demophoon* show a strong proximal expansion of the MFB pattern due to the development of red scales instead of black scales in this area of the wing (figure 3*a*, left; impacting 21.5 ± 11.3% of the forewing). In contrast, the *WntA* KOs of the co-mimic *H. m. rosina* showed a less pronounced proximal expansion (impacting 9.5 ± 3.9% of the forewing) and an observable distal expansion of the MFB pattern in several of the mutants (figure 3*a*, right; impacting 0.7 ± 0.1% of the forewing). By comparing the *WntA* mutant phenotypes for both species, we observed an area at the posterior of the distal margin of the MFB where *WntA* affects scale colouration in *H. e. demophoon* but not in *H. m. rosina* (figure 3*a*, orange triangles; covering 0.6 ± 0.1% of the forewing). This area overlaps with a forewing region in which wild-type individuals of *H. e. demophoon* and *H. m. rosina* show consistent difference (figure 2 and figure 3*b*, orange triangle). Moreover, MFB variation along the PC axis that strongly differentiated

pattern variation between *H. erato* and *H. melpomene* Postman populations (figure 1*d*, PC2) recapitulated the distal spot difference (figure 3*c*, orange triangle; covering 1.0 ± 0.2% of the forewing). In line with the observation by Concha *et al.* [12] of a distal expansion of the MFB in *H. m. rosina WntA* mutants, this PC axis also demonstrated a general distal margin expansion in *H. melpomene* Postman populations compared to *H. erato* populations (figure 3*c*, blue triangle; covering 2.9 ± 1.0% of the forewing).

## 4. Discussion

In *Heliconius* butterflies, convergence between co-mimicking populations has been broadly defined as nearly identical. In accordance with these phenotypic similarities, genetic work has suggested that convergence is governed by a small and shared set of genes [13,15,17,28,36]. For example, *WntA* has been linked to wing colour pattern variation in the forewing of both *H. erato* and *H. melpomene* butterflies [28]. Recently, the availability and implementation of CRISPR/Cas9 functional approaches in butterfly wing colour pattern development has allowed to further investigate the role of wing patterning genes [12,17,37]. One of the most striking results that emerged from the KO phenotypes is an apparent divergent developmental architecture underlying butterfly wing colour pattern convergence, despite *WntA* being involved in forewing band variation in both species. Hence, while the difference in KO phenotypes reflects distinct *WntA* wing colour pattern domains in the two divergent co-mimetic butterflies *H. e. demophoon* and *H. m. rosina* [12], it also highlights the power of natural selection in driving convergent adaptive phenotypes despite a divergent genetic and

developmental background. However, the extent to which divergence in *WntA* functioning may have affected the evolution of natural populations of mimetic phenotypes in *Heliconius* has not been investigated. By doing so, our study provides new insights into the dynamics of selection and how adaptation may be constrained to find alternative routes.

## (a) Patterns of advergence in mid-forewing band

A question that arises from the observed differences in MFB patterns between co-mimics is whether it is necessary for *H. erato* and *H. melpomene* to perfectly mimic each other and whether selective forces could work against convergence [19,38]. In our comparisons we observed that even though the position of forewing pattern elements may not be perfectly identical between co-mimics, the relative amount of black versus red or yellow is generally more similar than the absolute difference. This improved match of the size of the MFB seems to result from compensatory changes in the MFB pattern and are in line with phenotypic evolution being driven by predation pressure. A similar 'advergence' process has previously been demonstrated for the mimicry ring including *H. timareta thelxinoe*, *H. e. favorinus*, and *H. m. amaryllis* in Peru [26] and the mimicry ring including *H. e. phyllis*, *H. besckei*, *H. m. burchelli*, and *H. m. nanna* in Brazil [27]. They thus indicate fine scale pattern adaptation in non-homologous regions of the wing to obtain a better match in the shape and size of the pattern even though they have a shifted position in the wings. Differences in MFB patterns between sexes were the same for both species and suggest that sexual conflicts for species recognition are likely not driving species differences in MFB patterns.

The compensatory evolution to obtain a more similar area of the MFB despite its mismatch in position may suggest imperfect discrimination in the visual range of their predators, or the relative importance of overall features of colour contrast distribution rather than the exact position of pattern elements [20,21]. A remarkable example of this are the co-mimetic Ecuadorian butterflies *H. e. notabilis* and *H. m. plesseni* which both have red and white in their proximal forewing element but have the relative positions of white versus red colour inversed. Notably, these wing colour patterns are the results of complex epistatic interactions between *WntA* and other genes such as the transcription factor *optix*, which controls white and red scale development [13,36].

We also found detectable within-species differences between populations that are generally considered identical. Among Postman populations, the greatest MFB differences were observed between several *H. erato* populations (e.g. between *H. e. hydara* from Panama and French Guiana). Notably, these *H. erato* Postman populations also had the most apparent differences with its *H. melpomene* co-mimic (i.e. *H. e. hydara* versus *H. m. melpomene* from French Guiana). As *H. erato* is often suggested to be the more abundant co-mimic and, thus, the model which *H. melpomene* mimics [21,22], the presence of a less perfect mimetic signal in these cases may reflect a 'lag' in the evolution of better mimicry of the *H. melpomene* populations. This inference potentially fits a signal of recent adaptive evolution (i.e. selective sweep signal) across the regulatory regions in the first intronic region of *WntA* of *H. e. hydara* populations from French Guiana, but not *H. e. hydara* populations from Panama, or *H. m. melpomene* populations from French Guiana [25]. What drives the divergence in MFB pattern between the *H. erato* Postman populations may include local changes in the composition of the mimicry and/or predator community [18,24,26].

## (b) Indications of genetic and developmental constraints

Some of the pattern differences we observed between co-mimetic populations were shared among several geographically distinct populations and phenotypes. Such shared differences thus likely result from a shared genetic and developmental background of specific MFB areas among these geographical populations. For example, differences in MFB between *H. e. etylus* and *H. m. ecuadorensis* strongly resembled differences in the distal part of the 'Split' MFB of *H. e. notabilis* and *H. m. plesseni* ('S' in figure 2*a*). Similarly, the 'Broken' MFB phenotypes *H. e. erato* and *H. e. amalfreda* showed differences in the same areas of the wings compared to *H. m. thelxiopeia* and *H. m. meriana*, respectively ('B' in figure 2*a*). These pattern differences were observed regardless of interspecific wing shape differences, suggesting that differences in MFB phenotype between *H. erato* and *H. melpomene* are largely independent from shape differences in their wings.

Regarding the inter-species Postman population comparisons, our analyses showed a correlation between wild-type differences and differences in the *WntA* CRISPR/Cas9 KO phenotypes of *H. e. demophoon* and *H. m. rosina*. This correspondence between the *H. e. demophoon* and *H. m. rosina* *WntA* KO phenotypes and their wild-type observed mismatch was most obvious in the posterior spot at the distal margin of the MFB, where the *H. erato* Postman populations and KOs had red scales and the *H. melpomene* Postman populations and KOs always had black scales (figure 3, orange triangles). This observation highlights the existence of a *WntA* KO boundary and suggests that the observed imperfections in mimicry might be to some extent imposed by divergence in the gene regulatory network involved in the development of the MFB. As discussed earlier, it is possible that conflicting selection pressures can explain imperfections in mimicry. However, our quantitative wing colour pattern analyses on wild-type individuals and *WntA* KO phenotypes suggests the likely role of divergent genetic backgrounds in constraining the observed imperfect mimicry. While not obvious in the pairwise comparisons of the Postman co-mimics, the PCA analysis identified that the distal anterior margin of the MFB is generally expanded in *H. melpomene* compared to *H. erato*. Finally, the difference observed in the distal anterior margin of the MFB between co-mimicking Postman populations matched the distal boundary of *WntA* affected wing area as identified in the *H. e. demophoon* *WntA* KOs (figure 3*c*) and may potentially further highlight areas of the wing that are constrained by divergence in the gene regulatory network between *H. erato* and *H. melpomene*.

## (c) Candidates of divergence in the gene regulatory network

Divergence in the gene regulatory network that is involved in the development of the MFB may include a multitude of changes in upstream factors that regulate spatial and/or temporal expression of *WntA* [39], the *cis*-regulatory elements of *WntA* itself [29], as well as additional genes that define the fate of wing scale cells. A few of these diverged elements are likely loci or genes that have previously been implicated in

wing colour pattern variation in various *Heliconius* species. For example, the distal part of the MFB pattern that is expanded in *H. e. etylus*/*H. e. notabilis* compared to *H. m. ecuadorensis*/*H. m. plesseni* (figure 2) matches the distal area of the wing that is described to be affected by an additional locus called *Ro* [29,40–42]. Next, differences in the mismatch between the 'Split' MFB phenotypes *H. e. notabilis*/*H. m. plesseni* and *H. e. xenoclea*/*H. m. microclea* potentially result from epistatic interactions with the *optix* gene, affecting the size and shape of the MFB pattern. This has been previously suggested by looking at MFB patterns in hybrid butterflies that have the same *WntA* alleles but absence/presence of *optix* expression in the MFB [33]. Further, contrasting results have been found regarding the number of loci that affect MFB shape in *H. erato* compared to *H. melpomene*. For example, in *H. erato* the genetic architecture of MFB pattern variation has so far been mapped to the so-called *Sd*, *St*, and *Ly* loci that each affect a particular part of the MFB shape [40] and these loci have been demonstrated to include *cis*-regulatory elements of *WntA* [29]. In *H. melpomene*, on the other hand, regulatory variation at the so-called *N* locus, which includes the gene *cortex* and a few additional candidate genes, seems to underlie MFB shape variation together with *WntA* [40]. Potentially, this latter locus provides a candidate that explains the absence of a *WntA* effect on black wing scale development in the proximal area of the wing in *H. melpomene*. Finally, apart from the major effect loci involved in *Heliconius* colour patterns, quantitative trait loci (QTL) studies of pattern variation in *H. erato* and *H. melpomene* have demonstrated the existence of minor effect loci associated with quantitative changes in wing colour pattern [42–44]. This larger set of genetic variants controlling quantitative variation is additional to the regulatory complexity that modulates the expression of the major colour pattern genes [12,29,37].

## 5. Conclusion

Studying the evolutionary dynamics of adaptation is complicated due to the interplay of selection, genetics, and development. Adaptive radiations and mimicry systems have long been powerful study systems to unravel the genetic basis of adaptive traits. Here, we used *Heliconius* butterflies to investigate the interplay of selection and genetics and provide a tentative explanation on the causes that limit perfect wing mimicry after several million years of strong natural selection towards a mutual anti-predatory signal. We propose that phenotypic patterns of adaptation and convergence between *Heliconius* co-mimics is biased by divergence in the gene regulatory network underlying the mimicry trait. These networks can be highly polygenic and interconnected and evolution may not have been able to perfectly rewire them in the same identical way after the accumulation of changes across their genomes. In the case of the observed differences between the *WntA* mutant phenotypes of *H. erato* and *H. melpomene*, these constraints may exist due to independently evolved genetic elements that interact with the *WntA* gene or protein. These elements may include *trans* factors, *cis*-regulatory elements, or additional genes with a complementary role to *WntA* and may not have the genetic architecture to easily be detached from potential developmental interactions with other genes [45,46]. While we quantified exact wing colour pattern differences between *H. erato* and *H. melpomene* co-mimics, we also observed that the warning signal represented by the relative amount of colours is more consistent. Hence, changes to the MFB can be observed that compensate the developmental bias in other areas of the wing, indicating that selection has used an alternative route to convergence. Overall, our results provide a quantitative measure of the imperfection of mimicry and propose a novel view into the interplay between selective conflicts compared to genetic and developmental constraints in the production of similar phenotypes.

Data accessibility. All code to build the figures, landmarks, and images are available through GitHub repository (https://github.com/patternize-projects/Heliconius_forewing_band).

Competing interests. We declare we have no competing interests.

Funding. This work was funded by NSF EPSCoR RII Track-2 FEC (grant no. OIA 1736026) to R.P. and B.A.C. and NSF IOS 1656389 to R.P. S.M.V.B. was supported by a Puerto Rico Science, Technology & Research Trust catalyzer award (#2020-00142) and the NSF OIA 1736026. P.A.A.R. was supported by the Interdisciplinary & Quantitative Biology Summer Research Experience for Undergraduates (IQ BIO REU) Program funded by the National Science Foundation (grant no. DBI 1852259). H.C.G. was supported by The Puerto Rico Space Grant Consortium by NASA (grant no. NNX15AI11H).

Acknowledgements. We kindly thank Ananda Martins, Owen McMillan, Joe Hanly, Carolina Concha, Tim Thurman, Jennifer Cuthill, Markus Möst, Camilo Salazar, Mathieu Joron, Jake Morris, and Claire Mérot for sharing images and Chris Jiggins, Ian Warran, Patricio Salazar and Gabriela Montejo-Kovacevich for the earthcape repository (https://heliconius.ecdb.io). We thank Frederik Nijhout and Claire Mérot for thoughtful comments on the manuscript.

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
