## [Reviewer comments · Proceedings of the Royal Society B: Biological Sciences]

Review History

RSPB-2020-0063.R0 (Original submission)

Review form: Reviewer 1

Recommendation

Major revision is needed (please make suggestions in comments)

Scientific importance: Is the manuscript an original and important contribution to its field?

Excellent

General interest: Is the paper of sufficient general interest?

Excellent

Quality of the paper: Is the overall quality of the paper suitable?

Good

Is the length of the paper justified?

Yes

Should the paper be seen by a specialist statistical reviewer?

No

Do you have any concerns about statistical analyses in this paper? If so, please specify them explicitly in your report.

No

It is a condition of publication that authors make their supporting data, code and materials available - either as supplementary material or hosted in an external repository. Please rate, if applicable, the supporting data on the following criteria.

Is it accessible?

Yes

Is it clear?

Yes

Is it adequate?

Yes

Do you have any ethical concerns with this paper?

No

Comments to the Author

Review „Perfect mimicry between *Heliconius* butterflies is constrained by genetics and development“

The authors use image analysis of co-mimic *Heliconius* butterflies to assess variation in a specific colour pattern element and interpret their findings in the light of CRISPR/Cas9 knockout results published by Concha et al. (2019). They conclude that constraints in the gene regulatory network defining wing colour patterning underlies variation in colour pattern between comimics.

I think the question regarding the relevance and role of developmental constraints in adaptive evolution is topical and of broader interest and their approach and methodology are reasonable and lead to interesting results. My main criticism is that the manuscript, as it stands now, lacks clarity and precision. I'd appreciate a more detailed introduction to the question and the problem of developmental constraints already in the Introduction (they present examples in the Conclusion but I think they should be moved to the Introduction). In the Results section it is in places quite difficult to follow their arguments. On one hand, it is sometimes hard to link the details described in the text with the figures. The figures themselves are high quality and informative but some more precise references to the specifically discussed areas and boundaries would be useful (e.g. Fig 3). On the other hand, the argument why the comparison of the KO boundaries with the differences between the colour pattern regions in the two species is somewhat convoluted and should be presented in a clearer way - what/where exactly is the boundary mismatch, what can it tell us about the underlying mechanism and why is that. I don't think there is a flaw in their logic but the way it is presented could be improved substantially. Lastly, except for the Conclusion, the Discussion is very *Heliconius*-specific and I would suggest to discuss their findings in a broader context. In summary, I think the findings are interesting but need to be presented in a clearer way and I am sure the authors are able to deliver that.

Here are some detailed comments that illustrate my points raised above.

L27f: What exactly is meant by irreversible here - in a developmental context (a point of no return in development) or in the context of evolutionary change or both? This could be made clearer.

L53: „mid-forewing colour pattern shape“ is a quite long and rather cryptic term. It may require some explanation in the Intro, maybe a better introduction in the Abstract as well.

L63: a short paragraph describing the state of knowledge on how black scales are formed/controlled could be useful

L106-9: This bit may require a bit more precision. Readers won't necessarily know where WntA is expressed on the wing and where not?

L137: maybe a brief hint what the 90% quantile here means (e.g. by mentioning the number of KO

samples)?

L143: The problem of allometric differences is not presented clearly enough in my opinion. Is this a known problem, if so Ref. Adding the problematic landmarks in Figure S1B tension maps might help to understand why they were excluded?

L162-3 What is the full potential MFB? I assume this would be the full PCA space but this should be stated explicitly and maybe an example of a pattern that is not realized could be useful for understanding.

L166: naming one or several example population for the narrow median band could be useful

L166: "H. e notabilis" should be "H. e. notabilis"

L169-71: I think the statement that "widespread mismatch" is reflected by the PCAs in FigC&D is debatable - depending on the expectations one could also say the match is pretty good in many cases. Could help to statistically show the clusters are significantly different or adding a 95% ellipsoid

L173: maybe better "distal posterior" than "distal bottom"?

L184: I think it should be "ascribed to"

L188: "substantial" - I would not necessarily call these differences substantial. I also don't think this matters. The biological relevance or the thresholds that would affect bird behaviour are anyway unknown (or if not they should be mentioned) and for this paper (or the butterflies) it does not seem to matter whether we assess these differences as subtle or substantial. Maybe "detectable" would be neutral term

L200: remove extra dot in this sentence

L201 and L209: should be "thelxiopeia"

L198 and L202/3: the reasoning underlying these conclusions needs a bit more explanation (or should be moved to Discussion)

L217ff: this section is challenging - it's hard to follow the description and identifying the respective regions in Figure 3. I'd suggest improvement in precision and phrasing

L228-30: The arrows and description are hard to follow. I don't fully understand what the differently sized parentheses precisely refer to.

Figure 3 caption text: There is an asterisk in Fig 3B lower panel that is not explained (or I've missed it). One of the legends is ranging from 1 to 1 - shouldn't that be -1 to 1?

L233: the orange spot should be marked with e.g. an asterisk

L236-9: I struggled to understand this sentence. Is the distal boundary of the WntA mutants in H. e. demophoon shifted in comparison to the wildtype in this context here at all (except for the posterior orange spot)? And if the anterior distal boundary of H. e. demophoon WntA knockouts is identical to the wildtype why is this perfect match then interesting? In this context it may be helpful to add the green and yellow outlines to the first panel in Figure 3A or add a fourth panel with such an overlay to illustrate the exact differences between the KO and wildtype outline.

L285: what are the developmental WntA boundaries, where can they be seen...connect results with discussion

L307: a clearer reference to this "restricted area" would be desirable

L322: "greater imperfect mimicry" or maybe "less perfect"?

L325-6: Is developmental constraint the only explanation or could it just be time?

L269: should be "Divergence"

L358ff: I'd personally would prefer reading these examples presented here earlier in the Introduction or Discussion with a more general outline of the question.

Review form: Reviewer 2 (Claire Mérot)

Recommendation

Major revision is needed (please make suggestions in comments)

Scientific importance: Is the manuscript an original and important contribution to its field?

Excellent

General interest: Is the paper of sufficient general interest?

Excellent

Quality of the paper: Is the overall quality of the paper suitable?

Excellent

Is the length of the paper justified?

Yes

Should the paper be seen by a specialist statistical reviewer?

No

Do you have any concerns about statistical analyses in this paper? If so, please specify them explicitly in your report.

Yes

It is a condition of publication that authors make their supporting data, code and materials available - either as supplementary material or hosted in an external repository. Please rate, if applicable, the supporting data on the following criteria.

Is it accessible?

Yes

Is it clear?

Yes

Is it adequate?

Yes

Do you have any ethical concerns with this paper?

No

Comments to the Author

This manuscript uses an innovative approach to quantify colour pattern variation in a large dataset and map areas of resemblance and differences on the wing between co-mimetic species. Then, it links those results with the recent advances on the genetic basis of colour pattern regulation in *Heliconius*. Moreover, it analyses the phenotype in some KO mutants for a gene known to affect patterning and compares the portion of the wing affected by the mutation to differences quantified in wild butterflies. I found the analysis sound, the manuscript well-written and pleasant to read, the study original by its approach, accounting quantitatively for a trait that is too often only measured qualitatively (colour pattern). I think it is very interesting for geneticist, evolutionary biologist and ecologist. This manuscript opens a wide array of questions on the interplay between selection and genetic regulation with the compelling example of the evolution of mimicry.

That being said, and although I really like the manuscript, I am wondering to what extent some of the conclusions are not too far from what the data actually support. The manuscript makes several times the claim that they demonstrate developmental and genetic constraints, as a result of divergent gene regulatory network, but it seems to me rather an indirect evidence or an interpretation than a direct demonstration. Therefore I think that a substantial revision either toning down that claim, or providing stronger arguments for it, is desirable.

Please see my detailed comments and suggestions below.

- Major Comments:

#1-Concordance between data, analysis and general conclusion

The manuscript makes strong claims about the identification of developmental and genetic constraints:

L80 "Our data demonstrate the existence of species-specific developmental constraints that limit the ability of selection "

L368 "constraints in the convergence of phenotypes is here identified as the result of divergence in the gene regulatory network that interacts with the gene WntA"

Yet, when I look into the data, I see that the phenotypic approach allows to measure resemblance and to assess which portion of the wings remain different between species, and which converge but no direct link to genetics. The only analysis that relates to actual genetic compounds is limited to the analysis of few KO mutants for WntA. While this is novel and rare to see such kind of data, it seems that only a minor portion (distal) of the wing affected by the KO match differences observed between mimics in the wild. Therefore, my understanding is that most of the major claim comes from speculation about the position of colour pattern differences and position where WntA/other genes are known to be expressed. Or did I miss a point? (I'd be happy to be proven wrong since the conclusion are interesting).

Since the phenotypic data presented here is already impressive, one solution to avoid that speculative feeling and to better convinced the reader would be perhaps to re-focus the discussion on the data, while reducing the discussion about genetic regulation?

For instance, I think the major strength of the data is the repeatability of the 14 pairs. The fact that the same region of the wing cannot be blackened in any melpomene (cf results L173-175) despite variable selective pressure (different erato mimics) is actually one of the best arguments that there is a species-specific constraint in that area. The comparison between pairs is discussed extensively in the results (not always easy to follow but ok) but I missed this argument being brought in the discussion. I wonder also whether the consistency of some constraints could not appear more clearly on a figure. For now, one has to compare the column of differences with 14 wings on fig 2. Can we imagine one wing (or 1 for P, 1 for B, 1 for S) summarizing how often the same area represent a difference between erato/melpomene that goes in the same directions between pairs. (perhaps like Fig 3B for postman but with something indicating repeatability of mimicry differences: red in melpomene/black in erato in 3 pairs out of 5? In 5 pairs out of 5)??

Strength for the constraint conclusion also comes from detailing inter-individual variability vs co-mimic differences. It is contained in the present data but could be used as an argument if it were clearer in the results and used in the discussion. The intra-group variability in colour pattern is very important to defend the major point of the manuscript. In fact, the part of the colour pattern that vary within a population (inter-individual variability at boundaries for instance in a single subspecies), are more likely to be the regions under relaxed selection. By contrast what the region does not vary within a pop but vary consistently between co-mimetic species is rather due to constraints.

I hope the authors could reflect on intra-group variability and the power brought by the 14 pairs in the revised discussion to provide stronger arguments supporting the claim of the title "perfect mimicry is constrained by genetics and development"

#2 Intra-group variability

The dataset is good because the authors have 8-14 samples per population. Yet, I had difficulties following in the manuscript how the authors deal with inter-individual variability.

On PCA, for instance, this is really interesting because for some pairs of co-mimics it seems that the difference erato/mlepomene overlap with inter-individual variability (at least on PC1/PC2 for the blue hydara) while for other pairs, differences are quite consistent between all erato and

all co-mimetic Melpomene.

I see that such variability is taken into account in Fig 2 with the heatmap showing the proportion of individuals in which the pixel is coloured. Yet, the colour for the portion 0.2-0.5 are not well-visible (in particular the purple). Can the colour-scale be improved? It is a bit disappointing now to be unable to fully see inter-individual variability.

It must be that the differences between co-mimics illustrated on the last column of Fig 2 take that into account inter-individual variability but I can't figure out in the methods how the difference is calculated? Could this be better detailed in the methods?

Or is it just a difference of average? L118 "subtracting the average H. erato and H. Melpomene MFB pattern of each population" ? In such a case, it would be good to take into account intra-group variability.

#3 Clarify intra-sp vs inter-specific comparison

By essence the dataset is complex because there are multiply subspecies in two species, with different pairs of co-mimics, and sometimes the same pattern coming back in different geographical pairs. For the reader unfamiliar with the Heliconius system it may be hard to follow. I have tried to point places in which the authors should strive to clarify the distinction

L49-51 "positive frequency dependent selection imposed by birds has favour the evolution of over 25 geographically distinct mimicry populations"

I think that what the authors meant is that freq-dpdce selection has favoured mimicry?...

What has favoured the 25 geographically distinct rings has received several explanations (relaxed selection, hybridizations, colonization of new areas, etc) but freq-dpdce selection rather select against any novel patterns...

See for instance: Joron, M., & Mallet, J. L. (1998). Diversity in mimicry: paradox or paradigm?. *Trends in Ecology & Evolution*, 13(11), 461-466. (and more recent literature on the topic)

L54 "incredible diversity within each species yet qualitatively identical morphologies between each co-mimetic population" -> unclear that the second part of the sentence deals with inter-species mimicry.

L77-78 "all co-mimetics exhibit consistent differences in their forewing colour pattern shape" - ?? unclear where is the consistency? is it taking into account intra-pop variability or intra-subspecies variability (I mean across individuals of H; erato X, or across all pairs of erato/melpo)?

L188: "populations that are generally considered identical within H. erato" -> are we discussing here intra-specific variability?

#4 Robustness of the analysis of pattern based on landmarks

The matter to analyze pattern between mimics from different species is that vein landmarks tend to retain species-specific (and sex) information. The authors have addressed this matter by trying to keep the landmarks that do not recapitulate sex or species differences. I agree that the visualization of the PCA (Fig. S1) is quite convincing. Moreover, Fig S2-4 suggests that this matter of species/sex effect on landmarks does not affect the results. However, to confirm that there is no differences of shape between sexes or between species, one need to do a proper statistical test. What if the difference is on PC3?!

Please consider doing a manova to test for differences in shape between species or sex depending on the landmark subset. The number of PCs to enter into the Manova needs to be reduced probably.

See :

Vieira, V.M. 2012. Permutation tests to estimate significances on Principal Components Analysis. *Computational Ecology and Software* 2: 103-123.

Björklund, M. 2019. Be careful with your principal components. *Evolution* 73: 2151-2158.

5- About perfect mimicry

L40-41. I think that the concept of Mullerian mimicry deserves a better explanation for the unfamiliar reader since it is central to your point.

I was also surprised, given that the article is dealing with the matter of “perfect mimicry” not to see any ecological aspects on that point or paper on the evolution of mimicry. I know this is briefly mentioned in the last paragraph of discussion but it should be in introduction to assess whether selection is indeed expected to favour perfect mimicry (wouldn't coarse resemblance be enough??).

See perhaps some of those references.

Leimar, O., Tullberg, B.S. & Mallet, J. (2012) Mimicry, saltational evolution and the crossing of fitness valleys. *The Adaptive Landscape in Evolutionary Biology* (eds E.I. Svensson & R. Calsbeek), pp. 259–267. Oxford University Press, New York, NY, USA.

Ihalainen, E., Lindstrom, L., Mappes, J. & Puolakkainen, S. (2008) Can experienced birds select for Mullerian mimicry? *Behavioral Ecology*, 19, 362–368.

Ihalainen, E., Rowland, H.M., Speed, M.P., Ruxton, G.D. & Mappes, J. (2012) Prey community structure affects how predators select for Mullerian mimicry. *Proceedings of the Royal Society B-Biological Sciences*, 279, 2099–2105.

Penney, H.D., Hassall, C., Skevington, J.H., Abbott, K.R. & Sherratt, T.N. (2012) A comparative analysis of the evolution of imperfect mimicry. *Nature*, 483, 461–464.

Ruxton, G.D., Franks, D.W., Balogh, A.C.V. & Leimar, O. (2008) Evolutionary implications of the form of predator generalization for aposematic signals and mimicry in prey. *Evolution*, 62, 2913–2921

Moreover the results presented L206-209, that relative size of the colour area does not significantly vary between co-mimetic pairs is really interesting on the point of view of mimicry resemblance! Does this mean that when considering relative size (L207 “size” is it absolute or relative size?) variability between species is rarely more than variability between individuals. I think that this work on relative surface coloured brings another dimension to the paper that will raise interest not only from geneticist but also from ecologist. As discussed at the end of the discussion, that question whether the position or the accuracy of the patch is important or whether simply the global ratio colour/wing is not already good enough for mimicry. On the question genetic vs. selection, it means that, given the genetic constraints on some parts of the wings, other ways to enlarge the colour patch have emerged leading to convergence in surface... I wonder whether it would not be worth bringing back Fig S5 (or an improved version of it that takes into account intra-group variability into the main text?

#6 Figures are beautiful but the legend is hard to follow sometimes.

Fig. 1 I can't understand well the wing legends along the axis. I know from experience that such kind of variation is hard to represent... Perhaps simply trying to be more explicit by what “predicted” means?? Also since positive/negative image are a mirror of each other, do we need two wings per axis? + see my remark below on “expression of the pattern” = “presence of colour”!

Fig 2 same remark “proportion” below the 1st heat band could be “proportion of individuals with colour”?

- Minor comments:

L90 “Images were obtained through the authors' collections and collections made publicly available by Cuthill et al and Jiggins et al”

I couldn't help but noticing that some of the samples listed in Table S2 have been collected by myself (and M. Joron) and photos taken by myself... I am slightly surprised, but of course more than happy that those data are useful for more interesting studies. Perhaps the authors needs to

update this sentence or the acknowledgments? :-)

L194: "allometric". I am really puzzled by the use of this term throughout the manuscript. To me, and for all I could have read, allometry generally refers to relationship with size. Here, I don't see any analysis of size. Do the authors mean that the landmark excluded are the one that describe inter-specific differences of shape (regardless of pattern). Am I right? Please reconsider the use of the word "allometric", or if I am wrong, please define clearly this term for other readers; (in morphometrics it is always used for relationship with size).

Methods: how does the analysis takes into account the fuzziness of the border of the colour patch? Is there a threshold to draw the limit between colour and black? In my experience, *H. erato* has usually very sharp borders between the colour and the black while *H. melpomene* has a mix of red and black scales.

L126: "higher predicted expression of the pattern".

I think the term "expression" is misleading since it can be confounded by the region where a gene is expressed?? (cf L108 "focused on the area of the forewing in which WntA is not expressed"...?). My guess here is that it means "presence of colour" ? because what the authors look at is the position and size of a colour patch on a black background right?

I think the authors should aim for a more direct and simpler vocabulary. It could be "the colour pattern displayed "for L136 or "positive values represent a higher frequency in the presence of the colour"? for L126? Same in figure legends "proportion of the wing in which the trait is expressed". Wouldn't it be simpler for the reader "proportion of the wing that is coloured"?

Results part b: I suggest splitting the paragraph into sub paragraphs because as it stands now it is hard to follow. Perhaps explicitly split the results on the different kinds of co-mimics, then the results on surface/absolute difference? See how to integrate results on inter-individual variability?

L205" average absolute difference in the MFB pattern" -> please recall that this difference takes into account position.

L269 "Divergence" lack a "D"

Discussion

I feel that the second part of paragraph (a) and paragraph (b) have more to do with each other than the two paragraphs joined under title (a)...? Could they be joined and this would provide space for a 1st paragraph of the discussion more focused on the phenotypic data.

I suggest reorganizing the discussion to discuss (a) actual data on pattern, variability, consistency between races, why this leads to the conclusion of constraints for some areas, why not for others maybe (some boundaries seems to be as variable between species than within species). (b) gene regulatory network. (c) implication for mimicry evolution/selection, convergence via other roads toward same surface, etc...?

L263 "pattern similarity between co-mimetic *Heliconius* butterflies has been mostly described qualitatively"

□ But see: Mérot et al 2016 that the authors quote below.

Fig S1 : a typo on panel C, both axis are labelled PC2, while the x-axis is likely Pc1.

Suppl. Fig: please simplify vocabulary like in main figures.

Fig S5: it is unclear to me how absolute differences can be percentage?

Legend of Fig S3 and S4 are reversed in the text.

Thank you for this interesting manuscript,
Claire Mérot

Review form: Reviewer 3

Recommendation

Major revision is needed (please make suggestions in comments)

Scientific importance: Is the manuscript an original and important contribution to its field?

Acceptable

General interest: Is the paper of sufficient general interest?

Good

Quality of the paper: Is the overall quality of the paper suitable?

Acceptable

Is the length of the paper justified?

Yes

Should the paper be seen by a specialist statistical reviewer?

Yes

Do you have any concerns about statistical analyses in this paper? If so, please specify them explicitly in your report.

Yes

It is a condition of publication that authors make their supporting data, code and materials available - either as supplementary material or hosted in an external repository. Please rate, if applicable, the supporting data on the following criteria.

Is it accessible?

Yes

Is it clear?

Yes

Is it adequate?

Yes

Do you have any ethical concerns with this paper?

No

Comments to the Author

The article called "Perfect mimicry between *Heliconius* butterflies is constrained by genetics and development" evaluated interesting aspects from imperfect mimicry in *Heliconius* species be associates to genetic developmental constraints. The authors used landmarks and MFB to assess the wing pattern, and then, evaluated the relationship using WntA CRISPR/Cas9 KO phenotypes obtained by Concha et al. (2019). Finally, the authors found that genetic and developmental constraints avoid the perfect convergence to mimetic.

In my opinion, there are some points that should be carefully revised by the authors to improve the quality of this article, following below:

Major points

1) The weakest points of this article are related to the statistical analyses. Although the authors pointed that "Using quantitative measurements, we first show that all co-mimetics exhibit consistent differences in their mid-forewing colour pattern shapes" and "Comparing wing shape

between males and females showed no apparent difference in sex in both *H. erato* and *H. melpomene* (Figure S1C)." I do not believe the authors chose the best statistical analysis for evaluating the mimicry convergence and sex differences. The PCA analysis is important to visualize the patterns, working as an exploratory analysis. Although, the authors should run statistical analysis appropriate to assess the difference in wing pattern evaluated in this article. Please see Rossato et al (2018), this article has a similar approach and could give some ways to conduct the statistical analyses for multivariate data to assess mimicry convergence and compare male to female wings.

2) It was not clear which statistical analysis was used for evaluating the MFB differences between co-mimics included in results (lines 204-215). Please include the statistical approach in the materials section.

4) Although the authors said they used a quantitative approach, it seems that they based the interpretation in the qualitative description in: "Mismatch between co-mimics coincides with developmental WntA boundaries".

5) I was expecting that the last paragraph from the introduction included hypotheses associated with the objectives. However, the authors pointed out the conclusions found in the article.

6) At the results section, the authors discussed the results (for example, lines 212-213 and 236-239). I think it is better to keep this to the discussion section.

7) I suggest the authors review the discussion after running the statistical analysis. I believe the interpretation of some of the results could change.

8) Please, give a background about genetic architecture associated with phenotype, since it was not completely clear in the present article.

9) The results found using the distinct data sets are a little bit disconnected. Please, make sure they are linked to the same issue. Then, connect the interpretation between wing aspects measured using landmark and MFB to the data set using CRISPR phenotype.

10) Please include in the discussion others selective forces that could work in distinct directions against the convergent evolution, as proposed by Estrada and Jiggins (2008), and, discuss the results found by (Rossato et al 2018) according to the imperfect convergence in *Heliconius* co-mimetic and sexual selection.

Minor points

Line 146 and (2) and - exclude one "and"

Line 269 - Include D to Divergence

References

Estrada, C., and Jiggins, C. D. 2008. Interspecific sexual attraction because of convergence in warning coloration: Is there a conflict between natural and sexual selection in mimetic species? *Journal of Evolutionary Biology*, 28, 749-760.

Rossato, D. O., D. Boligon, R. Fornel, M. R. Kronforst, G. L. Gonçalves, and G. R. P. Moreira. 2018. Subtle variation in size and shape of the whole forewing and the red band among co-mimics revealed by geometric morphometric analysis in *Heliconius* butterflies. *Ecology and Evolution*.

Decision letter (RSPB-2020-0063.R0)

18-Feb-2020

Dear Dr Van Belleghem:

I am writing to inform you that your manuscript RSPB-2020-0063 entitled "Perfect mimicry between *Heliconius* butterflies is constrained by genetics and development" has, in its current form, been rejected for publication in *Proceedings B*.

This action has been taken on the advice of referees, who have recommended that substantial revisions are necessary. With this in mind we would be happy to consider a resubmission, provided the comments of the referees are fully addressed. However please note that this is not a provisional acceptance.

Sincerely,
 Professor Loeske Kruuk
 mailto: proceedingsb@royalsociety.org

Associate Editor
 Board Member: 1
 Comments to Author:

All three reviewers are quite enthusiastic about the manuscript, however all three have identified important areas that need to be improved before it can be considered further. I also very much like the manuscript. Reviewer 1 and 2 raise numerous helpful points about clarity and interpretation that should be address. Reviewer 3 also raises an issue about the statistical methods that the authors should consider.

Reviewer(s)' Comments to Author:

Referee: 1

Comments to the Author(s)

Review „Perfect mimicry between *Heliconius* butterflies is constrained by genetics and development“

The authors use image analysis of co-mimic *Heliconius* butterflies to assess variation in a specific colour pattern element and interpret their findings in the light of CRISPR/Cas9 knockout results published by Concha et al. (2019). They conclude that constraints in the gene regulatory network defining wing colour patterning underlies variation in colour pattern between comimics.

I think the question regarding the relevance and role of developmental constraints in adaptive evolution is topical and of broader interest and their approach and methodology are reasonable and lead to interesting results. My main criticism is that the manuscript, as it stands now, lacks

clarity and precision. I'd appreciate a more detailed introduction to the question and the problem of developmental constraints already in the Introduction (they present examples in the Conclusion but I think they should be moved to the Introduction). In the Results section it is in places quite difficult to follow their arguments. On one hand, it is sometimes hard to link the details described in the text with the figures. The figures themselves are high quality and informative but some more precise references to the specifically discussed areas and boundaries would be useful (e.g. Fig 3). On the other hand, the argument why the comparison of the KO boundaries with the differences between the colour pattern regions in the two species is somewhat convoluted and should be presented in a clearer way – what/where exactly is the boundary mismatch, what can it tell us about the underlying mechanism and why is that. I don't think there is a flaw in their logic but the way it is presented could be improved substantially. Lastly, except for the Conclusion, the Discussion is very *Heliconius*-specific and I would suggest to discuss their findings in a broader context. In summary, I think the findings are interesting but need to be presented in a clearer way and I am sure the authors are able to deliver that.

Here are some detailed comments that illustrate my points raised above.

L27f: What exactly is meant by irreversible here – in a developmental context (a point of no return in development) or in the context of evolutionary change or both? This could be made clearer.

L53: „mid-forewing colour pattern shape“ is a quite long and rather cryptic term. It may require some explanation in the Intro, maybe a better introduction in the Abstract as well.

L63: a short paragraph describing the state of knowledge on how black scales are formed/controlled could be useful

L106-9: This bit may require a bit more precision. Readers won't necessarily know where WntA is expressed on the wing and where not?

L137: maybe a brief hint what the 90% quantile here means (e.g. by mentioning the number of KO samples)?

L143: The problem of allometric differences is not presented clearly enough in my opinion. Is this a known problem, if so Ref. Adding the problematic landmarks in Figure S1B tension maps might help to understand why they were excluded?

L162-3 What is the full potential MFB? I assume this would be the full PCA space but this should be stated explicitly and maybe an example of a pattern that is not realized could be useful for understanding.

L166: naming one or several example population for the narrow median band could be useful

L166: “*H. e notabilis*” should be “*H. e. notabilis*”

L169-71: I think the statement that “widespread mismatch” is reflected by the PCAs in FigC&D is debatable – depending on the expectations one could also say the match is pretty good in many cases. Could help to statistically show the clusters are significantly different or adding a 95% ellipsoid

L173: maybe better “distal posterior” than “distal bottom”?

L184: I think it should be “ascribed to”

L188: “substantial” – I would not necessarily call these differences substantial. I also don't think this matters. The biological relevance or the thresholds that would affect bird behaviour are anyway unknown (or if not they should be mentioned) and for this paper (or the butterflies) it does not seem to matter whether we assess these differences as subtle or substantial. Maybe “detectable” would be neutral term

L200: remove extra dot in this sentence

L201 and L209: should be “*thelxiopeia*”

L198 and L202/3: the reasoning underlying these conclusions needs a bit more explanation (or should be moved to Discussion)

L217ff: this section is challenging – it's hard to follow the description and identifying the respective regions in Figure 3. I'd suggest improvement in precision and phrasing

L228-30: The arrows and description are hard to follow. I don't fully understand what the differently sized parentheses precisely refer to.

Figure 3 caption text: There is an asterisk in Fig 3B lower panel that is not explained (or I've missed it). One of the legends is ranging from 1 to 1 – shouldn't that be -1 to 1?

L233: the orange spot should be marked with e.g. an asterisk
 L236-9: I struggled to understand this sentence. Is the distal boundary of the WntA mutants in *H. e. demophoon* shifted in comparison to the wildtype in this context here at all (except for the posterior orange spot)? And if the anterior distal boundary of *H. e. demophoon* WntA knockouts is identical to the wildtype why is this perfect match then interesting? In this context it may be helpful to add the green and yellow outlines to the first panel in Figure 3A or add a fourth panel with such an overlay to illustrate the exact differences between the KO and wildtype outline.
 L285: what are the developmental WntA boundaries, where can they be seen...connect results with discussion
 L307: a clearer reference to this "restricted area" would be desirable
 L322: "greater imperfect mimicry" or maybe "less perfect"?
 L325-6: Is developmental constraint the only explanation or could it just be time?
 L269: should be "Divergence"
 L358ff: I'd personally would prefer reading these examples presented here earlier in the Introduction or Discussion with a more general outline of the question.

Referee: 2

Comments to the Author(s)

This manuscript uses an innovative approach to quantify colour pattern variation in a large dataset and map areas of resemblance and differences on the wing between co-mimetic species. Then, it links those results with the recent advances on the genetic basis of colour pattern regulation in *Heliconius*. Moreover, it analyses the phenotype in some KO mutants for a gene known to affect patterning and compares the portion of the wing affected by the mutation to differences quantified in wild butterflies. I found the analysis sound, the manuscript well-written and pleasant to read, the study original by its approach, accounting quantitatively for a trait that is too often only measured qualitatively (colour pattern). I think it is very interesting for geneticist, evolutionary biologist and ecologist. This manuscript opens a wide array of questions on the interplay between selection and genetic regulation with the compelling example of the evolution of mimicry.

That being said, and although I really like the manuscript, I am wondering to what extent some of the conclusions are not too far from what the data actually support. The manuscript makes several times the claim that they demonstrate developmental and genetic constraints, as a result of divergent gene regulatory network, but it seems to me rather an indirect evidence or an interpretation than a direct demonstration. Therefore I think that a substantial revision either toning down that claim, or providing stronger arguments for it, is desirable.

Please see my detailed comments and suggestions below.

- Major Comments:

#1-Concordance between data, analysis and general conclusion

The manuscript makes strong claims about the identification of developmental and genetic constraints:

L80 "Our data demonstrate the existence of species-specific developmental constraints that limit the ability of selection "

L368 "constraints in the convergence of phenotypes is here identified as the result of divergence in the gene regulatory network that interacts with the gene WntA"

Yet, when I look into the data, I see that the phenotypic approach allows to measure resemblance and to assess which portion of the wings remain different between species, and which converge but no direct link to genetics. The only analysis that relates to actual genetic compounds is limited to the analysis of few KO mutants for WntA. While this is novel and rare to see such kind of data, it seems that only a minor portion (distal) of the wing affected by the KO match differences

observed between mimics in the wild. Therefore, my understanding is that most of the major claim comes from speculation about the position of colour pattern differences and position where *WntA*/other genes are known to be expressed. Or did I miss a point? (I'd be happy to be proven wrong since the conclusion are interesting).

Since the phenotypic data presented here is already impressive, one solution to avoid that speculative feeling and to better convinced the reader would be perhaps to re-focus the discussion on the data, while reducing the discussion about genetic regulation?

For instance, I think the major strength of the data is the repeatability of the 14 pairs. The fact that the same region of the wing cannot be blackened in any melpomene (cf results L173-175) despite variable selective pressure (different erato mimics) is actually one of the best arguments that there is a species-specific constraint in that area. The comparison between pairs is discussed extensively in the results (not always easy to follow but ok) but I missed this argument being brought in the discussion. I wonder also whether the consistency of some constraints could not appear more clearly on a figure. For now, one has to compare the column of differences with 14 wings on fig 2. Can we imagine one wing (or 1 for P, 1 for B, 1 for S) summarizing how often the same area represent a difference between erato/melpomene that goes in the same directions between pairs. (perhaps like Fig 3B for postman but with something indicating repeatability of mimicry differences: red in melpomene/black in erato in 3 pairs out of 5? In 5 pairs out of 5)??

Strength for the constraint conclusion also comes from detailing inter-individual variability vs co-mimic differences. It is contained in the present data but could be used as an argument if it were clearer in the results and used in the discussion. The intra-group variability in colour pattern is very important to defend the major point of the manuscript. In fact, the part of the colour pattern that vary within a population (inter-individual variability at boundaries for instance in a single subspecies), are more likely to be the regions under relaxed selection. By contrast what the region does not vary within a pop but vary consistently between co-mimetic species is rather due to constraints.

I hope the authors could reflect on intra-group variability and the power brought by the 14 pairs in the revised discussion to provide stronger arguments supporting the claim of the title "perfect mimicry is constrained by genetics and development"

#2 Intra-group variability

The dataset is good because the authors have 8-14 samples per population. Yet, I had difficulties following in the manuscript how the authors deal with inter-individual variability.

On PCA, for instance, this is really interesting because for some pairs of co-mimics it seems that the difference erato/mlepomene overlap with inter-individual variability (at least on PC1/PC2 for the blue hy dara) while for other pairs, differences are quite consistent between all erato and all co-mimetic Melpomene.

I see that such variability is taken into account in Fig 2 with the heatmap showing the proportion of individuals in which the pixel is coloured. Yet, the colour for the portion 0.2-0.5 are not well-visible (in particular the purple). Can the colour-scale be improved? It is a bit disappointing now to be unable to fully see inter-individual variability.

It must be that the differences between co-mimics illustrated on the last column of Fig 2 take that into account inter-individual variability but I can't figure out in the methods how the difference is calculated? Could this be better detailed in the methods?

Or is it just a difference of average? L118 "subtracting the average H. erato and H. Melpomene MFB pattern of each population" ? In such a case, it would be good to take into account intra-group variability.

#3 Clarify intra-sp vs inter-specific comparison

By essence the dataset is complex because there are multiply subspecies in two species, with different pairs of co-mimics, and sometimes the same pattern coming back in different geographical pairs. For the reader unfamiliar with the *Heliconius* system it may be hard to follow. I have tried to point places in which the authors should strive to clarify the distinction

L49-51 “positive frequency dependent selection imposed by birds has favour the evolution of over 25 geographically distinct mimicry populations”

I think that what the authors meant is that freq-dpdce selection has favoured mimicry?...

What has favoured the 25 geographically distinct rings has received several explanations (relaxed selection, hybridizations, colonization of new areas, etc) but freq-dpdce selection rather select against any novel patterns...

See for instance: Joron, M., & Mallet, J. L. (1998). Diversity in mimicry: paradox or paradigm?. *Trends in Ecology & Evolution*, 13(11), 461-466. (and more recent literature on the topic)

L54 “incredible diversity within each species yet qualitatively identical morphologies between each co-mimetic population” -> unclear that the second part of the sentence deals with inter-species mimicry.

L77-78 “all co-mimetics exhibit consistent differences in their forewing colour pattern shape” - ?? unclear where is the consistency? is it taking into account intra-pop variability or intra-subspecies variability (I mean across individuals of *H. erato* X, or across all pairs of *erato/melpo*)?

L188: “populations that are generally considered identical within *H. erato*” -> are we discussing here intra-specific variability?

#4 Robustness of the analysis of pattern based on landmarks

The matter to analyze pattern between mimics from different species is that vein landmarks tend to retain species-specific (and sex) information. The authors have addressed this matter by trying to keep the landmarks that do not recapitulate sex or species differences. I agree that the visualization of the PCA (Fig. S1) is quite convincing. Moreover, Fig S2-4 suggests that this matter of species/sex effect on landmarks does not affect the results. However, to confirm that there is no differences of shape between sexes or between species, one need to do a proper statistical test. What if the difference is on PC3?!

Please consider doing a manova to test for differences in shape between species or sex depending on the landmark subset. The number of PCs to enter into the Manova needs to be reduced probably.

See :

Vieira, V.M. 2012. Permutation tests to estimate significances on Principal Components Analysis. *Computational Ecology and Software* 2: 103-123.

Björklund, M. 2019. Be careful with your principal components. *Evolution* 73: 2151-2158.

5- About perfect mimicry

L40-41. I think that the concept of Mullerian mimicry deserves a better explanation for the unfamiliar reader since it is central to your point.

I was also surprised, given that the article is dealing with the matter of “perfect mimicry” not to see any ecological aspects on that point or paper on the evolution of mimicry. I know this is briefly mentioned in the last paragraph of discussion but it should be in introduction to assess whether selection is indeed expected to favour perfect mimicry (wouldn't coarse resemblance be enough??).

See perhaps some of those references.

Leimar, O., Tullberg, B.S. & Mallet, J. (2012) Mimicry, saltational evolution and the crossing of fitness valleys. *The Adaptive Landscape in Evolutionary Biology* (eds E.I. Svensson & R. Calsbeek), pp. 259–267. Oxford University Press, New York, NY, USA.

Ihalainen, E., Lindstrom, L., Mappes, J. & Puolakkainen, S. (2008) Can experienced birds select for Mullerian mimicry? *Behavioral Ecology*, 19, 362–368.

Ihalainen, E., Rowland, H.M., Speed, M.P., Ruxton, G.D. & Mappes, J. (2012) Prey community structure affects how predators select for Mullerian mimicry. *Proceedings of the Royal Society B-Biological Sciences*, 279, 2099–2105.

Penney, H.D., Hassall, C., Skevington, J.H., Abbott, K.R. & Sherratt, T.N. (2012) A comparative analysis of the evolution of imperfect mimicry. *Nature*, 483, 461–464.

Ruxton, G.D., Franks, D.W., Balogh, A.C.V. & Leimar, O. (2008) Evolutionary implications of the form of predator generalization for aposematic signals and mimicry in prey. *Evolution*, 62, 2913–2921

Moreover the results presented L206-209, that relative size of the colour area does not significantly vary between co-mimetic pairs is really interesting on the point of view of mimicry resemblance! Does this mean that when considering relative size (L207 “size” is it absolute or relative size?) variability between species is rarely more than variability between individuals. I think that this work on relative surface coloured brings another dimension to the paper that will raise interest not only from geneticist but also from ecologist. As discussed at the end of the discussion, that question whether the position or the accuracy of the patch is important or whether simply the global ratio colour/wing is not already good enough for mimicry. On the question genetic vs. selection, it means that, given the genetic constraints on some parts of the wings, other ways to enlarge the colour patch have emerged leading to convergence in surface... I wonder whether it would not be worth bringing back Fig S5 (or an improved version of it that takes into account intra-group variability into the main text?)

#6 Figures are beautiful but the legend is hard to follow sometimes.

Fig. 1 I can't understand well the wing legends along the axis. I know from experience that such kind of variation is hard to represent... Perhaps simply trying to be more explicit by what “predicted” means?? Also since positive/negative image are a mirror of each other, do we need two wings per axis? + see my remark below on “expression of the pattern” = “presence of colour”!

Fig 2 same remark “proportion” below the 1st heat band could be “proportion of individuals with colour”?

• Minor comments:

L90 “Images were obtained through the authors’ collections and collections made publicly available by Cuthill et al and Jiggins et al”

I couldn't help but noticing that some of the samples listed in Table S2 have been collected by myself (and M. Joron) and photos taken by myself... I am slightly surprised, but of course more than happy that those data are useful for more interesting studies. Perhaps the authors needs to update this sentence or the acknowledgments? :-)

L94: “allometric”. I am really puzzled by the use of this term throughout the manuscript. To me, and for all I could have read, allometry generally refers to relationship with size. Here, I don't see any analysis of size. Do the authors mean that the landmark excluded are the one that describe inter-specific differences of shape (regardless of pattern). Am I right? Please reconsider the use of the word “allometric”, or if I am wrong, please define clearly this term for other readers; (in morphometrics it is always used for relationship with size).

Methods: how does the analysis takes into account the fuzziness of the border of the colour patch? Is there a threshold to draw the limit between colour and black? In my experience, *H. erato* has usually very sharp borders between the colour and the black while *H. melpomene* has a mix of red and black scales.

L126: “higher predicted expression of the pattern”.

I think the term “expression” is misleading since it can be confounded by the region where a gene is expressed?? (cf L108 “focused on the area of the forewing in which WntA is not expressed”...??). My guess here is that it means “presence of colour” ? because what the authors look at is the position and size of a colour patch on a black background right?

I think the authors should aim for a more direct and simpler vocabulary. It could be “the colour pattern displayed “for L136 or “positive values represent a higher frequency in the presence of the colour”? for L126? Same in figure legends “proportion of the wing in which the trait is expressed”. Wouldn't it be simpler for the reader “proportion of the wing that is coloured”?

Results part b: I suggest splitting the paragraph into sub paragraphs because as it stands now it is hard to follow. Perhaps explicitly split the results on the different kinds of co-mimics, then the results on surface/absolute difference? See how to integrate results on inter-individual variability?

L205 “average absolute difference in the MFB pattern” -> please recall that this difference takes into account position.

L269 “Divergence” lack a “D”

Discussion

I feel that the second part of paragraph (a) and paragraph (b) have more to do with each other than the two paragraphs joined under title (a)...? Could they be joined and this would provide space for a 1st paragraph of the discussion more focused on the phenotypic data.

I suggest reorganizing the discussion to discuss (a) actual data on pattern, variability, consistency between races, why this leads to the conclusion of constraints for some areas, why not for others maybe (some boundaries seems to be as variable between species than within species). (b) gene regulatory network. (c) implication for mimicry evolution/selection, convergence via other roads toward same surface, etc...?

L263 “pattern similarity between co-mimetic *Heliconius* butterflies has been mostly described qualitatively”

□ But see: Mérot et al 2016 that the authors quote below.

Fig S1 : a typo on panel C, both axis are labelled PC2, while the x-axis is likely Pc1.
Suppl. Fig: please simplify vocabulary like in main figures.

Fig S5: it is unclear to me how absolute differences can be percentage?
Legend of Fig S3 and S4 are reversed in the text.

Thank you for this interesting manuscript,
Claire Mérot

Referee: 3

Comments to the Author(s)

The article called "Perfect mimicry between *Heliconius* butterflies is constrained by genetics and development" evaluated interesting aspects from imperfect mimicry in *Heliconius* species by associates to genetic developmental constraints. The authors used landmarks and MFB to assess the wing pattern, and then, evaluated the relationship using WntA CRISPR/Cas9 KO phenotypes obtained by Concha et al. (2019). Finally, the authors found that genetic and developmental constraints avoid the perfect convergence to mimetic.

In my opinion, there are some points that should be carefully revised by the authors to improve the quality of this article, following below:

Major points

- 1) The weakest points of this article are related to the statistical analyses. Although the authors pointed that "Using quantitative measurements, we first show that all co-mimetics exhibit consistent differences in their mid-forewing colour pattern shapes" and "Comparing wing shape between males and females showed no apparent difference in sex in both *H. erato* and *H. melpomene* (Figure S1C)." I do not believe the authors chose the best statistical analysis for evaluating the mimicry convergence and sex differences. The PCA analysis is important to visualize the patterns, working as an exploratory analysis. Although, the authors should run statistical analysis appropriate to assess the difference in wing pattern evaluated in this article. Please see Rossato et al (2018), this article has a similar approach and could give some ways to conduct the statistical analyses for multivariate data to assess mimicry convergence and compare male to female wings.
- 2) It was not clear which statistical analysis was used for evaluating the MFB differences between co-mimics included in results (lines 204-215). Please include the statistical approach in the materials section.
- 4) Although the authors said they used a quantitative approach, it seems that they based the interpretation in the qualitative description in: "Mismatch between co-mimics coincides with developmental WntA boundaries".
- 5) I was expecting that the last paragraph from the introduction included hypotheses associated with the objectives. However, the authors pointed out the conclusions found in the article.
- 6) At the results section, the authors discussed the results (for example, lines 212-213 and 236-239). I think it is better to keep this to the discussion section.
- 7) I suggest the authors review the discussion after running the statistical analysis. I believe the interpretation of some of the results could change.
- 8) Please, give a background about genetic architecture associated with phenotype, since it was not completely clear in the present article.
- 9) The results found using the distinct data sets are a little bit disconnected. Please, make sure they are linked to the same issue. Then, connect the interpretation between wing aspects measured using landmark and MFB to the data set using CRISPR phenotype.
- 10) Please include in the discussion others selective forces that could work in distinct directions against the convergent evolution, as proposed by Estrada and Jiggins (2008), and, discuss the results found by (Rossato et al 2018) according to the imperfect convergence in *Heliconius* co-mimetic and sexual selection.

Minor points

Line 146 and (2) and - exclude one "and"

Line 269 - Include D to Divergence

References

Estrada, C., and Jiggins, C. D. 2008. Interspecific sexual attraction because of convergence in warning coloration: Is there a conflict between natural and sexual selection in mimetic species? *Journal of Evolutionary Biology*, 28, 749–760.

Rossato, D. O., D. Boligon, R. Fornel, M. R. Kronforst, G. L. Gonçalves, and G. R. P. Moreira. 2018. Subtle variation in size and shape of the whole forewing and the red band among co-mimics revealed by geometric morphometric analysis in *Heliconius* butterflies. *Ecology and Evolution*.

Author's Response to Decision Letter for (RSPB-2020-0063.R0)

See Appendix A.

RSPB-2020-1267.R0

Review form: Reviewer 1

Recommendation

Accept with minor revision (please list in comments)

Scientific importance: Is the manuscript an original and important contribution to its field?

Excellent

General interest: Is the paper of sufficient general interest?

Excellent

Quality of the paper: Is the overall quality of the paper suitable?

Excellent

Is the length of the paper justified?

Yes

Should the paper be seen by a specialist statistical reviewer?

No

Do you have any concerns about statistical analyses in this paper? If so, please specify them explicitly in your report.

No

It is a condition of publication that authors make their supporting data, code and materials available - either as supplementary material or hosted in an external repository. Please rate, if applicable, the supporting data on the following criteria.

Is it accessible?

Yes

Is it clear?

Yes

Is it adequate?

Yes

Do you have any ethical concerns with this paper?

No

Comments to the Author

Van Belleghem et al. have revised this manuscript and it is now considerably improved, much clearer and more accessible. They also answered all my questions. The authors could maybe slightly improve and extend their explanation of their argument that the overlap of KO areas and wild-type differences between species suggests the presence of different developmental networks. It's a key point in this MS and sufficiently explained but a slightly more elaborate explanation may be helpful for a readers less familiar with this topic.

I also have a few additional minor comments:

L22: „define“ should be „defines“ or „network“ should be „networks“

L59: better „genes“?

L64: „relaxed selection due to coarse discrimination by predators“ – important point but I am not sure relaxed selection falls into the category „conflicting selection pressures“?

L90: I think both divergence in cis-elements as well as trans-elements affect epistatic interaction (e.g. trans-elements bind to cis-elements) but the line seems to suggest that epistatic interactions are always trans?

L166: states „between each of the co-mimicking pairs“ but the subsequently presented statistics and results are for „between the species“, I think

L297: e.

Review form: Reviewer 2 (Claire Mérot)

Recommendation

Accept with minor revision (please list in comments)

Scientific importance: Is the manuscript an original and important contribution to its field?

Excellent

General interest: Is the paper of sufficient general interest?

Excellent

Quality of the paper: Is the overall quality of the paper suitable?

Excellent

Is the length of the paper justified?

Yes

Should the paper be seen by a specialist statistical reviewer?

No

Do you have any concerns about statistical analyses in this paper? If so, please specify them explicitly in your report.

No

It is a condition of publication that authors make their supporting data, code and materials available - either as supplementary material or hosted in an external repository. Please rate, if applicable, the supporting data on the following criteria.

Is it accessible?

Yes

Is it clear?

Yes

Is it adequate?

Yes

Do you have any ethical concerns with this paper?

No

Comments to the Author

This is a revised version of an interesting manuscript that I have reviewed before, and I am still very enthusiastic about its relevance and interest.

The authors have nicely and deeply improved the manuscript, which stands stronger and much easier to read. They have addressed all of my concerns appropriately and, to the extent of my knowledge, this is also true for other's reviewers' remarks.

I made below a couple of suggestions to improve clarity but those are really minor.
Claire Mérot

- Minor points

Abstract: I feel that the abstract is long and vague about the context (particularly L14-17) without highlighting the exciting results (same area repeatedly divergent between pairs of mimics - overlap with KO boundaries)... Perhaps some re-writing to have more impact?

L68: It is really strange to have "here, we aim.." and then again "L93 "in the current study, we aim". I'd removed L68-70 to rather jump into presenting the system and keep L93 the exact scope of the study

L85: "a textbook example of Mullerian mimicry" -& either already above or comes a bit late? Perhaps this whole part could be L68 to introduce the system and the question.

L114 "describe" -& described

L144: "5 mutants"? how many per species?

L168: what about other races than postman?

L173: Why "as for wing shape"? -& not mentioned before

L175: 55% or 58% of classification is indeed low. To reject more robustly any strong effect of sex, it would be good to show that 55% of classification is in fact similar to classification between random groups (this could be done just by resampling two groups of similar sample size as male and female but mixing the sex label, and ask what % of classification is obtained. I suspect this would be a range of 40-60% meaning that in fact the sex effect on pattern is unlikely to affect the results)

L179-181: cool result! And it is now nicely visible with the orange triangle

L202: I wonder whether this sentence about the control of consistent results with a diff set of landmarks cannot go into supplementary or methods to streamline the results?

L238 "One of the most striking results that emerged from the KO phenotypes is the divergent developmental architecture underlying butterfly wing color pattern convergence"
-& I don't get that at all. The KO confirm that the same gene (WntA) controls the same trait in both species (when look broadly). That looks like a similar (conserved? Or convergent?) architecture to me ... Should make it easy to be mimics... But then the exact domains affected by this gene does vary -& hence the question of whether divergence at WntA or related network or background effect explain imperfect mimicry. Perhaps be more precise about the scale at which there is convergence and divergence?

Fig 2: nicely improved!! However, it would be appreciated to have both the limits of the erato KO and the Melpomene KO on the last column. This difference is at the core of the study and the main conclusions.

Review form: Reviewer 3

Recommendation

Major revision is needed (please make suggestions in comments)

Scientific importance: Is the manuscript an original and important contribution to its field?

Good

General interest: Is the paper of sufficient general interest?

Excellent

Quality of the paper: Is the overall quality of the paper suitable?

Good

Is the length of the paper justified?

Yes

Should the paper be seen by a specialist statistical reviewer?

Yes

Do you have any concerns about statistical analyses in this paper? If so, please specify them explicitly in your report.

Yes

It is a condition of publication that authors make their supporting data, code and materials available - either as supplementary material or hosted in an external repository. Please rate, if applicable, the supporting data on the following criteria.

Is it accessible?

Yes

Is it clear?

Yes

Is it adequate?

Yes

Do you have any ethical concerns with this paper?

No

Comments to the Author

The article intituled "Perfect mimicry between Heliconius butterflies is constrained by genetics and development" assessed the convergence in wing pattern between co-mimics using innovative and sophisticate tools. I am convinced that this work will contribute to many areas of knowledge, mainly to ecological and evolutionary studies. However, there are some aspects that I believe could be improved.

The main point is related to the connection between the parts of the article. I don't know if I missed, unfortunately, but the connection between the first part (Color pattern analysis) and the second part (WntA CRISPR KO analysis) of the article is not completely clear for me. I understood both separately, but I can not see how the WntA CRISPR KO phenotype help to explain the imperfection in wing colour pattern in 13 mimicking races. Is there some selective pressure selecting the pattern in the same way between all co-mimetics? Additionally, I did not understand how the inclusion of the WntA CRISPR KO phenotype of *Heliconius erato* demophon in Figure 2 contribute to understanding the pattern of the co-mimetic, and why the

authors choose the phenotype of *Heliconius erato demophoon* and not of *Heliconius melpomene rosina* to represent the mutant pattern.

- I suggest a change in the expression "the size of the MFB" for "the value of the MFB", avoiding the confusion with the allometric relationship between size and shape.

- In lines 165 -167, if I have understood correctly, the posterior probability of classification was assessed in *H. erato* and *H. melpomene*; however, I believe the posterior probability could be assessed within population (co-mimetics). Make more sense according to the objectives of the article.

- Please, include degrees of freedom in results from statistic F.

- In lines 174-175, the authors did not find a high posterior probability of classification in sexes; however, as many other works suggest there is a high difference between male and female, I would like to know if the authors assessed the difference between male and female within species.

Decision letter (RSPB-2020-1267.R0)

26-Jun-2020

Dear Dr Van Belleghem,

Your manuscript has now been peer reviewed and the reviews have been assessed by an Associate Editor. The reviewers' comments (not including confidential comments to the Editor) and the comments from the Associate Editor are included at the end of this email for your reference. As you will see, the reviewers and the Editors have raised some concerns with your manuscript and we would like to invite you to revise your manuscript to address them.

Research ethics:

Use of animals and field studies:

Please submit a copy of your revised paper within three weeks. If we do not hear from you within this time your manuscript will be rejected. If you are unable to meet this deadline (especially given the current circumstances), please let us know as soon as possible, as we may be able to grant an extension.

Thank you for submitting your manuscript to Proceedings B; we look forward to receiving your revision. If you have any questions at all, please do not hesitate to get in touch. Finally, I hope you and your co-authors are well in these challenging times.

Best wishes,
 Professor Loeske Kruuk
 Editor
 mailto: proceedingsb@royalsociety.org

Associate Editor Board Member
 Comments to Author:

All three original reviewers found the manuscript to be much improved, and all are very enthusiastic about the study. However, all three made numerous suggestions to improve the clarity and presentation of the manuscript. I do not expect that these suggested changes will be very difficult to do, as there are no additional analyses required. I look forward to seeing a revision!

Reviewer(s)' Comments to Author:

Referee: 2

Comments to the Author(s).

This is a revised version of an interesting manuscript that I have reviewed before, and I am still very enthusiastic about its relevance and interest.

The authors have nicely and deeply improved the manuscript, which stands stronger and much easier to read. They have addressed all of my concerns appropriately and, to the extent of my knowledge, this is also true for other's reviewers' remarks.

I made below a couple of suggestions to improve clarity but those are really minor.
 Claire Mérot

- Minor points

Abstract: I feel that the abstract is long and vague about the context (particularly L14-17) without highlighting the exciting results (same area repeatedly divergent between pairs of mimics - overlap with KO boundaries)... Perhaps some re-writing to have more impact?

L68: It is really strange to have "here, we aim.." and then again "L93 "in the current study, we aim". I'd removed L68-70 to rather jump into presenting the system and keep L93 the exact scope of the study

L85: "a textbook example of Mullerian mimicry" -> either already above or comes a bit late? Perhaps this whole part could be L68 to introduce the system and the question.

L114 "describe" -> described

L144: "5 mutants"? how many per species?

L168: what about other races than postman?

L173: Why "as for wing shape"? -> not mentioned before

L175: 55% or 58% of classification is indeed low. To reject more robustly any strong effect of sex, it would be good to show that 55% of classification is in fact similar to classification between random groups (this could be done just by resampling two groups of similar sample size as male

and female but mixing the sex label, and ask what % of classification is obtained. I suspect this would be a range of 40-60% meaning that in fact the sex effect on pattern is unlikely to affect the results)

L179-181: cool result! And it is now nicely visible with the orange triangle

L202: I wonder whether this sentence about the control of consistent results with a diff set of landmarks cannot go into supplementary or methods to streamline the results?

L238 "One of the most striking results that emerged from the KO phenotypes is the divergent developmental architecture underlying butterfly wing color pattern convergence"

-> I don't get that at all. The KO confirm that the same gene (WntA) controls the same trait in both species (when look broadly). That looks like a similar (conserved? Or convergent?) architecture to me ... Should make it easy to be mimics... But then the exact domains affected by this gene does vary -> hence the question of whether divergence at WntA or related network or background effect explain imperfect mimicry. Perhaps be more precise about the scale at which there is convergence and divergence?

Fig 2: nicely improved!! However, it would be appreciated to have both the limits of the erato KO and the Melpomene KO on the last column. This difference is at the core of the study and the main conclusions.

Referee: 1

Comments to the Author(s).

Van Belleghem et al. have revised this manuscript and it is now considerably improved, much clearer and more accessible. They also answered all my questions. The authors could maybe slightly improve and extend their explanation of their argument that the overlap of KO areas and wild-type differences between species suggests the presence of different developmental networks. It's a key point in this MS and sufficiently explained but a slightly more elaborate explanation may be helpful for a readers less familiar with this topic.

I also have a few additional minor comments:

L22: „define“ should be „defines“ or „network“ should be „networks“

L59: better „genes“?

L64: „relaxed selection due to coarse discrimination by predators“ – important point but I am not sure relaxed selection falls into the category „conflicting selection pressures“?

L90: I think both divergence in cis-elements as well as trans-elements affect epistatic interaction (e.g. trans-elements bind to cis-elements) but the line seems to suggest that epistatic interactions are always trans?

L166: states „between each of the co-mimicking pairs“ but the subsequently presented statistics and results are for „between the species“, I think

L297: e.

Referee: 3

Comments to the Author(s).

The article intituled "Perfect mimicry between Heliconius butterflies is constrained by genetics and development" assessed the convergence in wing pattern between co-mimics using innovative and sophisticate tools. I am convinced that this work will contribute to many areas of knowledge, mainly to ecological and evolutionary studies. However, there are some aspects that I believe could be improved.

The main point is related to the connection between the parts of the article. I don't know if I missed, unfortunately, but the connection between the first part (Color pattern analysis) and the

second part (WntA CRISPR KO analysis) of the article is not completely clear for me. I understood both separately, but I can not see how the WntA CRISPR KO phenotype help to explain the imperfection in wing colour pattern in 13 mimicking races. Is there some selective pressure selecting the pattern in the same way between all co-mimetics? Additionally, I did not understand how the inclusion of the WntA CRISPR KO phenotype of *Heliconius erato* demophon in Figure 2 contribute to understanding the pattern of the co-mimetic, and why the authors choose the phenotype of *Heliconius erato* demophon and no of *Heliconius melpomene* rosina to represent the mutant pattern.

- I suggest a change in the expression "the size of the MFB" for "the value of the MFB", avoiding the confusion with the allometric relationship between size and shape.

- In lines 165 -167, if I have understood correctly, the posterior probability of classification was assessed in *H. erato* and *H. melpomene*; however, I believe the posterior probability could be assessed within population (co-mimetics). Make more sense according to the objectives of the article.

- Please, include degrees of freedom in results from statistic F.

- In lines 174-175, the authors did not find a high posterior probability of classification in sexes; however, as many other works suggest there is a high difference between male and female, I would like to know if the authors assessed the difference between male and female within species.

Author's Response to Decision Letter for (RSPB-2020-1267.R0)

See Appendix B.

Decision letter (RSPB-2020-1267.R1)

30-Jun-2020

Dear Dr Van Belleghem,

I am pleased to inform you that your manuscript entitled "Perfect mimicry between *Heliconius* butterflies is constrained by genetics and development" has been accepted for publication in *Proceedings B*.

Open Access

Paper charges

Thank you for submitting your paper to the Proceedings B. On behalf of all the Editors, we look forward to your continued contributions to the Journal.

Yours sincerely,
Professor Loeske Kruuk
Editor, Proceedings B
mailto: proceedingsb@royalsociety.org

Associate Editor:

Board Member

Comments to Author:

The authors have done an excellent job responding to the relatively minor comments of the reviewers. I am happy to recommend that the paper be accepted at Proceedings B, and look forward to seeing it in print!

Appendix A

RESPONSES

18-Feb-2020

Dear Dr Van Belleghem:

I am writing to inform you that your manuscript RSPB-2020-0063 entitled "Perfect mimicry between Heliconius butterflies is constrained by genetics and development" has, in its current form, been rejected for publication in Proceedings B.

This action has been taken on the advice of referees, who have recommended that substantial revisions are necessary. With this in mind we would be happy to consider a resubmission, provided the comments of the referees are fully addressed. However please note that this is not a provisional acceptance.

Sincerely,

Professor Loeske Kruuk
mailto:proceedingsb@royalsociety.org

Associate Editor
Board Member: 1
Comments to Author:

All three reviewers are quite enthusiastic about the manuscript, however all three have identified important areas that need to be improved before it can be considered further. I also very much like the manuscript. Reviewer 1 and 2 raise numerous helpful points about clarity and interpretation that should be address. Reviewer 3 also raises an issue about the statistical methods that the authors should consider.

Reviewer(s)' Comments to Author:

Referee: 1

Comments to the Author(s)

Review „Perfect mimicry between Heliconius butterflies is constrained by genetics and development“
The authors use image analysis of co-mimic Heliconius butterflies to assess variation in a specific colour pattern element and interpret their findings in the light of CRISPR/Cas9 knockout results published by Concha et al. (2019). They conclude that constraints in the gene regulatory network defining wing colour patterning underlies variation in colour pattern between comimics.

I think the question regarding the relevance and role of developmental constraints in adaptive evolution is topical and of broader interest and their approach and methodology are reasonable and lead to interesting results. My main criticism is that the manuscript, as it stands now, lacks clarity and precision. I'd appreciate a more detailed introduction to the question and the problem of developmental constraints already in the Introduction (they present examples in the Conclusion but I think they should be moved to the Introduction). In the Results section it is in places quite difficult to follow their arguments. On one hand, it is sometimes hard to link the details described in the text with the figures. The figures themselves are high quality and informative but some more precise references to the specifically discussed areas and boundaries would be useful (e.g. Fig 3). On the other hand, the argument why the comparison of the KO boundaries with the differences between the colour pattern regions in the two species is somewhat convoluted and should be presented in a clearer way – what/where exactly is the boundary mismatch, what can it tell us about the underlying mechanism and why is that. I don't think there is a flaw in their logic but the way it is presented could be improved substantially. Lastly, except for the Conclusion, the Discussion is very Heliconius-specific and I would suggest to discuss their findings in a broader context. In summary, I think the findings are interesting but need to be presented in a clearer way and I am sure the authors are able to deliver that.

>> Thank you for your thoughtful comments and suggestions, which we believe have helped us tremendously to improve our manuscript.

We improved the introduction to our questions and hypothesis as follows:

First, we moved examples from the conclusion to the introduction, which, as the reviewer kindly notes, adds to the broader scope of our findings.

Second, we have improved our last paragraph to better introduce our hypothesis as follows:

- “In the current study, we aim to explore the contribution of genetic and developmental constraints on the phenotypic mismatch between comimetic pairs of *Heliconius* butterflies. We therefore first quantitatively measured and compared differences in the mid-forewing band (MFB) pattern of 14 co-mimicking populations of the butterfly species *H. erato* and *H. melpomene* from Central and South America. We then interpret these differences in light of the recent *Heliconius WntA* CRISPR/Cas9 KO mutants [12]. We were specifically interested to test how species-specific divergence might impact the developmental function of *WntA* and limit adaptive convergence. Our results suggest the possible existence of constraints, imposed by genetic and developmental differences, for natural selection to achieve perfect mimicry. By taking advantage of the most recent methodological and technological advances provided by functional experiments and computational quantitative measures, our study offers an alternative approach to artificial selection experiments to test the relative constraint of genetics and development to adaptation.”

Third, we improved the link between the figures and the details described in the text. We resolved this issue by adding the outline of the *H. e. demophoon WntA* CRISPR KO pattern to the differences between *H. erato* and *H. melpomene* races. We think this helps comparing the different races within species and how all the other races, apart from the postman, may relate to the *WntA* KO's.

Finally, we recognize that due to word limits the discussion is still largely *Heliconius*-specific, but we now better describe how our results relate to patterns of advergence and constraints. The broader context is elaborated mainly in the introduction.

Here are some detailed comments that illustrate my points raised above.

L27: What exactly is meant by irreversible here – in a developmental context (a point of no return in development) or in the context of evolutionary change or both? This could be made clearer.

>> This was meant as a point of no return in the context of evolutionary change. We changed the statement to:

- “In this regard, key developmental steps can limit or bias trait variation, posing so-called constraints on the directionality of evolution [2–4]. Consequently, when populations evolve independently, lineages can accumulate key changes that lead evolution along an irreversible trajectory [5]. Understanding the relative contribution (or constraint) of genetics and development to adaptation would therefore allow us to better comprehend the directionality and predictability of evolution [6].”

L53: „mid-forewing colour pattern shape” is a quite long and rather cryptic term. It may require some explanation in the Intro, maybe a better introduction in the Abstract as well.

>> We removed this sentence in the introduction and introduce the mid-forewing band later. For simplicity, we changed mid-forewing to forewing in the abstract, which should not change the interpretation.

L63: a short paragraph describing the state of knowledge on how black scales are formed/controlled could be useful.

>> We added the following:

- “*WntA* is a member of the Wnt family of signaling ligands and a key molecular tool for butterfly wing color pattern development. While the gene coding sequence is highly conserved across lepidoptera, its *cis*-regulatory diversity underlies wing pattern shape variation between and within butterflies species [28,29]. Recent CRISPR/Cas9 KO experiments have shown its role in defining the ultimate color fate of individual wing scale cells and highlighted incredible variability in the position and wing territory affected by *WntA* in *H. erato* and *H. melpomene* [12].”

L106-9: This bit may require a bit more precision. Readers won't necessarily know where WntA is expressed on the wing and where not?

>> We changed this part to:

- “While *WntA* is involved in black scale development in both the fore- and hindwing in *Heliconius* [12,16], differences in the distribution of *WntA* have mainly been found to correlate with the position of black color in the central part of the forewing among *Heliconius* races [33]. With the interest of studying variation in the MFB, we extracted and focused on the area of the forewing in which black is absent and in which *WntA* is thus likely not expressed.”

L137: maybe a brief hint what the 90% quantile here means (e.g. by mentioning the number of KO samples)?

>> We changed and moved this to the legend of Figure 3:

- “The yellow and green outlines show the *WntA* KO area as present in at least 2 mutant butterflies.”

L143: The problem of allometric differences is not presented clearly enough in my opinion. Is this a known problem, if so Ref. Adding the problematic landmarks in Figure S1B tension maps might help to understand why they were excluded?

>> As reviewer 2 correctly noted, we misused the term allometry to mean interspecific wing shape differences. We now also added the reference to Merot et al. 2016 and Rossato et al. 2018 to this statement, as they previously identified species difference in wing shape. Additionally, we added the problematic landmarks to Figure S1B.

L162-3 What is the full potential MFB? I assume this would be the full PCA space but this should be stated explicitly and maybe an example of a pattern that is not realized could be useful for understanding.

>> We removed this statement.

L166: naming one or several example population for the narrow median band could be useful

>> We changed the sentence to:

- “The second main axis of variation (PC2) in the MFB shape is dominated by the presence of either a narrow median band, as observed in *H. e. cyrbia/H. m. cythera*, *H. e. lativitta/H. m. malleti* and *H. e. emma/H. m. aglaope*, or two spots, as observed in the *H. e. notabilis/H. m. plesseni* and *H. e. microclea/H. m. xenoclea* populations (Figure 1C).”

L166: “H. e notabilis” should be “H. e. notabilis”

>> Fixed

L169-71: I think the statement that “widespread mismatch” is reflected by the PCAs in FigC&D is debatable – depending on the expectations one could also say the match is pretty good in many cases. Could help to statistically show the clusters are significantly different or adding a 95% ellipsoid

>> We agree and removed the word “widespread”. We have also now performed appropriate statistical tests to support these and other statements. For both the landmark analysis and the wing color patterns, we have now tested the effect of population, sex and species on shape variables by using multivariate analysis of variance (MANOVA) as implemented in R v3.5. For this, we used only the values of samples along significant PC axes as determined by the *permutationPA* function in the R package *jackstraw* (as advised by reviewer 2). Shape discrimination between *H. erato* and *H. melpomene* and posterior probability of classification was studied using linear discriminant analysis (LDA) as implemented in the R package MASS.

L173: maybe better “distal posterior” than “distal bottom”?

>> Changed to “distal posterior”.

L184: I think it should be “ascribed to”

>> Removed

L188: “substantial” – I would not necessarily call these differences substantial. I also don’t think this matters. The biological relevance or the thresholds that would affect bird behaviour are anyway unknown (or if not they should be mentioned) and for this paper (or the butterflies) it does not seem to

matter whether we assess these differences as subtle or substantial. Maybe “detectable” would be neutral term

>> We agree and changed “substantial divergence” to “detectable within-species differences”.

L200: remove extra dot in this sentence

>> Fixed

L201 and L209: should be “thelxiopeia”

>> Fixed

L198 and L202/3: the reasoning underlying these conclusions needs a bit more explanation (or should be moved to Discussion)

>> We moved these statements to the discussion and improved the context in which they are stated.

L217ff: this section is challenging – it’s hard to follow the description and identifying the respective regions in Figure 3. I’d suggest improvement in precision and phrasing

>> We significantly shortened this section by removing the more speculative discussion points and kept it focused on the main differences between the mutant phenotypes and how they match to wild type differences between the co-mimics. We also added better annotations to the figures with a descriptive link in the text.

L228-30: The arrows and description are hard to follow. I don’t fully understand what the differently sized parentheses precisely refer to.

>> We replaced these arrows in Figure 3 with differently colored triangles. They are meant to indicate areas that differ between the *H. e. demophoon* and *H. m. rosina* mutants and also match areas that differ between the postman phenotype mimics.

Figure 3 caption text: There is an asterisk in Fig 3B lower panel that is not explained (or I’ve missed it). One of the legends is ranging from 1 to 1 – shouldn’t that be -1 to 1?

>> We removed the asterisk and fixed the legend.

L233: the orange spot should be marked with e.g. an asterisk

>> Now marked with an orange triangle.

L236-9: I struggled to understand this sentence. Is the distal boundary of the WntA mutants in *H. e. demophoon* shifted in comparison to the wildtype in this context here at all (except for the posterior

orange spot)? And if the anterior distal boundary of *H. e. demophoon* *WntA* knockouts is identical to the wildtype why is this perfect match then interesting? In this context it may be helpful to add the green and yellow outlines to the first panel in Figure 3A or add a fourth panel with such an overlay to illustrate the exact differences between the KO and wildtype outline.

>> Apart from the distal/posterior 'orange' spot, the expansion of the distal area of *H. e. rosina* compared to *H. e. demophoon* is indeed minimal and we now more appropriately mention this observation. However, from the PCA analysis on only postman phenotypes, a large part of the variation between melpomene and erato includes an expansion of the distal margin in melpomene postman phenotypes compared to erato postman phenotypes. This difference matches the difference observed between *H. e. demophoon WntA* KO's and the *H. m. rosina WntA* KO's.

We added better comparisons of the wildtype and mutant phenotypes to Figure 3.

L285: what are the developmental *WntA* boundaries, where can they be seen...connect results with discussion

>> We added the following to improve the link with the results:

- “Regarding the inter-species Postman race comparisons, our analyses showed a correlation between wild-type differences and differences in the *WntA* CRISPR/Cas9 KO phenotypes of *H. e. demophoon* and *H. m. rosina*. This correspondence between the *H. e. demophoon* and *H. m. rosina WntA* KO phenotypes and their wild-type observed mismatch was most obvious in the posterior spot at the distal margin of the MFB, where the *H. erato* Postman races and KO's had red scales and the *H. melpomene* Postman races and KO's always had black scales (Figure 3, orange triangles). This observation highlights the existence of a *WntA* KO boundary and suggests that the observed imperfections in mimicry might be to some extent imposed by divergence in the gene regulatory network involved in the development of the MFB.”

L307: a clearer reference to this “restricted area” would be desirable

>> We changed this sentence to:

- “this latter locus provides a strong candidate that explains the absence of a *WntA* effect on black wing scale development in the proximal area of the wing in *H. melpomene*.”

L322: “greater imperfect mimicry” or maybe “less perfect”?

>> Changed to “less perfect”.

L325-6: Is developmental constraint the only explanation or could it just be time?

>> We now moved this statement into the general discussion of differences observed between *H. erato* and *H. melpomene*, before the discussion of potential genetic and developmental constraints. We

indeed agree that time could explain the mismatch between the co-mimics (i.e. “lag”), but genetic constraints could increase the time needed to converge better.

L269: should be “Divergence”

>> Fixed

L358ff: I’d personally would prefer reading these examples presented here earlier in the Introduction or Discussion with a more general outline of the question.

>> We moved these examples to the introduction and improved the presentation of our question and their impact.

Referee: 2

Comments to the Author(s)

This manuscript uses an innovative approach to quantify colour pattern variation in a large dataset and map areas of resemblance and differences on the wing between co-mimetic species. Then, it links those results with the recent advances on the genetic basis of colour pattern regulation in *Heliconius*. Moreover, it analyses the phenotype in some KO mutants for a gene known to affect patterning and compares the portion of the wing affected by the mutation to differences quantified in wild butterflies. I found the analysis sound, the manuscript well-written and pleasant to read, the study original by its approach, accounting quantitatively for a trait that is too often only measured qualitatively (colour pattern). I think it is very interesting for geneticist, evolutionary biologist and ecologist. This manuscript opens a wide array of questions on the interplay between selection and genetic regulation with the compelling example of the evolution of mimicry.

>> Thank you!

That being said, and although I really like the manuscript, I am wondering to what extent some of the conclusions are not too far from what the data actually support. The manuscript makes several times the claim that they demonstrate developmental and genetic constraints, as a result of divergent gene regulatory network, but it seems to me rather an indirect evidence or an interpretation than a direct demonstration. Therefore, I think that a substantial revision either toning down that claim, or providing stronger arguments for it, is desirable.

>> We agree that the species differences in *WntA* KO boundaries may not solely explain the differences in mid-forewing band pattern observed between the co-mimics, as relaxed and sexual selection may contribute to these differences. However, we believe that the finding of natural species differences and species differences in *WntA* KO boundaries coinciding is a strong indicator of the contribution of developmental constraints to these differences. In general agreement with this comment, we have replaced the word “demonstrate” with “suggest” in all instances regarding this interpretation.

Please see my detailed comments and suggestions below.

- Major Comments:

#1-Concordance between data, analysis and general conclusion

The manuscript makes strong claims about the identification of developmental and genetic constraints: L80 “Our data demonstrate the existence of species-specific developmental constraints that limit the ability of selection “

>> Changed to:

- “Our results suggest the possible existence of constraints, imposed by genetic and developmental differences, for natural selection to achieve perfect mimicry.”

L368 “constraints in the convergence of phenotypes is here identified as the result of divergence in the gene regulatory network that interacts with the gene *WntA*”

>> Changed to:

- “This observation highlights the existence of a *WntA* KO boundary and suggests that the observed imperfections in mimicry might be to some extent imposed by divergence in the gene regulatory network involved in the development of the MFB.”

Yet, when I look into the data, I see that the phenotypic approach allows to measure resemblance and to assess which portion of the wings remain different between species, and which converge but no direct link to genetics. The only analysis that relates to actual genetic compounds is limited to the analysis of few KO mutants for *WntA*. While this is novel and rare to see such kind of data, it seems that only a minor portion (distal) of the wing affected by the KO match differences observed between mimics in the wild. Therefore, my understanding is that most of the major claim comes from speculation about the position of colour pattern differences and position where *WntA*/other genes are known to be expressed. Or did I miss a point? (I’d be happy to be proven wrong since the conclusion are interesting).

Since the phenotypic data presented here is already impressive, one solution to avoid that speculative feeling and to better convince the reader would be perhaps to re-focus the discussion on the data, while reducing the discussion about genetic regulation?

>> We agree with the reviewer and refocused the introduction and discussion on the data by reducing speculation on the regulation or the potential genetic architecture underlying the phenotypic differences between *H. erato* and *H. melpomene*. Additionally, we give a more balanced interpretation of the phenotypic differences in light of potential selective conflicts that may also explain mismatches in phenotype (see below # 5- About perfect mimicry).

For instance, I think the major strength of the data is the repeatability of the 14 pairs. The fact that the same region of the wing cannot be blackened in any melpomene (cf results L173-175) despite variable selective pressure (different erato mimics) is actually one of the best arguments that there is a species-specific constraint in that area. The comparison between pairs is discussed extensively in the results (not always easy to follow but ok) but I missed this argument being brought in the discussion.

>> We streamlined these results and connected their description better to the figures. The section “(b) Indications of genetic and developmental constraints” now brings forward this argument in the discussion.

I wonder also whether the consistency of some constraints could not appear more clearly on a figure. For now, one has to compare the column of differences with 14 wings on fig 2. Can we imagine one wing (or 1 for P, 1 for B, 1 for S) summarizing how often the same area represent a difference between erato/melpomene that goes in the same directions between pairs. (perhaps like Fig 3B for postman but

with something indicating repeatability of mimicry differences: red in melpomene/black in erato in 3 pairs out of 5? In 5 pairs out of 5)??

>> We attempted to resolve this issue by adding the outline of the *H. e. demophoon* WntA CRISPR KO pattern to the differences between *H. erato* and *H. melpomene* races. We think this helps comparing the different races within species and how all the other races, apart from the postman, may relate to the WntA KO's. Presenting the repeatability of the mimicry difference generally is quite difficult, because of the large differences between the MFB phenotypes between geographic races, even between some of the *H. erato* postman races. However, this variability is very interesting in itself, as it may suggest that the gene regulation of WntA may have starkly diverged among these populations.

Strength for the constraint conclusion also comes from detailing inter-individual variability vs co-mimic differences. It is contained in the present data but could be used as an argument if it were clearer in the results and used in the discussion. The intra-group variability in colour pattern is very important to defend the major point of the manuscript. In fact, the part of the colour pattern that vary within a population (inter-individual variability at boundaries for instance in a single subspecies), are more likely to be the regions under relaxed selection. By contrast what region does not vary within a pop but vary consistently between co-mimetic species is rather due to constraints.

I hope the authors could reflect on intra-group variability and the power brought by the 14 pairs in the revised discussion to provide stronger arguments supporting the claim of the title "perfect mimicry is constrained by genetics and development"

>> We believe our results have now been solidified by performing more appropriate statistical assessment of the intra and inter group variability. First, the inter-species differences between co-mimics are found to be highly significant when controlling for the intra group variability (modified figure 2). Second, we provide a more through discussion of the patterns of mismatch between the 14 co-mimics in the discussion (section (b) Indications of genetic and developmental constraints).

#2 Intra-group variability

The dataset is good because the authors have 8-14 samples per population. Yet, I had difficulties following in the manuscript how the authors deal with inter-individual variability.

On PCA, for instance, this is really interesting because for some pairs of co-mimics it seems that the difference erato/melpomene overlap with inter-individual variability (at least on PC1/PC2 for the blue hydara) while for other pairs, differences are quite consistent between all erato and all co-mimetic Melpomene.

I see that such variability is taken into account in Fig 2 with the heatmap showing the proportion of individuals in which the pixel is coloured. Yet, the colour for the portion 0.2-0.5 are not well-visible (in particular the purple). Can the colour-scale be improved? It is a bit disappointing now to be unable to fully see inter-individual variability.

>> We agree the scale was hard to read in this range of values. The color scale in the heatmaps of the intra-species variability was particularly chosen to be color-blind friendly using the R package *viridis*. We now changed this color palette to brewers blue, which we think gives a better visibility in the lower range. However, we hope this issue is also resolved by (1) better statistical comparisons of the PCA results using MANOVA and (2) the third column in Fig 2 highlighting the parts in the wing that are different between all samples included in the pairwise comparisons (i.e. 100% difference is either fully red or blue depending on which species the trait is absent or present).

It must be that the differences between co-mimics illustrated on the last column of Fig 2 take that into account inter-individual variability but I can't figure out in the methods how the difference is calculated? Could this be better detailed in the methods?

Or is it just a difference of average? L118 "subtracting the average *H. erato* and *H. Melpomene* MFB pattern of each population" ? In such a case, it would be good to take into account intra-group variability.

>> The illustration in the third column of figure 2 is indeed the difference of the average frequency of the pattern in *erato* vs. *melpomene* in each pixel. We have modified the sentence in the methods as follows:

- "Differences in the MFB patterns were first compared by subtracting the *H. erato* and *H. melpomene* pattern frequencies of each population, obtained with the *sumRaster* function in *patternize* (i.e. absolute MFB difference) and compared between co-mimics using a one-sample t-test".

#3 Clarify intra-sp vs inter-specific comparison

By essence the dataset is complex because there are multiply subspecies in two species, with different pairs of co-mimics, and sometimes the same pattern coming back in different geographical pairs. For the reader unfamiliar with the *Heliconius* system it may be hard to follow. I have tried to point places in which the authors should strive to clarify the distinction

L49-51 "positive frequency dependent selection imposed by birds has favoured the evolution of over 25 geographically distinct mimicry populations"

I think that what the authors meant is that freq-dpdce selection has favoured mimicry?...

What has favoured the 25 geographically distinct rings has received several explanations (relaxed selection, hybridizations, colonization of new areas, etc) but freq-dpdce selection rather select against any novel patterns...

See for instance: Joron, M., & Mallet, J. L. (1998). Diversity in mimicry: paradox or paradigm?. *Trends in Ecology & Evolution*, 13(11), 461-466. (and more recent literature on the topic)

>> We agree the sentence was misleading and changed it to:

- “This phenotypic convergence has evolved through strong selection pressures that benefit a common warning pattern that birds have learned to associate with unpalatability.”

L54 “incredible diversity within each species yet qualitatively identical morphologies between each co-mimetic population” -> unclear that the second part of the sentence deals with inter-species mimicry.

>> We removed this sentence in the introduction as a response to a comment by reviewer 1 and because it is unnecessary.

L77-78 “all co-mimetics exhibit consistent differences in their forewing colour pattern shape” – ?? unclear where is the consistency? is it taking into account intra-pop variability or intra-subspecies variability (I mean across individuals of *H. erato* X, or across all pairs of *erato/melpo*)?

>> We removed this statement from the introduction.

L188: “populations that are generally considered identical within *H. erato*” -> are we discussing here intra-specific variability?

>> Yes. We changed this sentence to:

- “We also found detectable within-species differences between populations that are generally considered identical.”

#4 Robustness of the analysis of pattern based on landmarks

The matter to analyze pattern between mimics from different species is that vein landmarks tend to retain species-specific (and sex) information. The authors have addressed this matter by trying to keep the landmarks that do not recapitulate sex or species differences. I agree that the visualization of the PCA (Fig. S1) is quite convincing. Moreover, Fig S2-4 suggests that this matter of species/sex effect on landmarks does not affect the results. However, to confirm that there is no differences of shape between sexes or between species, one need to do a proper statistical test. What if the difference is on PC3?!

Please consider doing a manova to test for differences in shape between species or sex depending on the landmark subset. The number of PCs to enter into the Manova needs to be reduced probably.

See :

Vieira, V.M. 2012. Permutation tests to estimate significances on Principal Components Analysis. *Computational Ecology and Software* 2: 103-123.

Björklund, M. 2019. Be careful with your principal components. *Evolution* 73: 2151-2158.

>> For both the landmark analysis and the wing color patterns, we have now tested the effect of population, sex and species on shape variables by using multivariate analysis of variance (MANOVA) as implemented in R v3.5. For this, we used only the values of samples along significant PC axes as determined by the *permutationPA* function in the R package *jackstraw* as suggested by Vieire 2012 and Bjorklund 2019. Shape discrimination between *H. erato* and *H. melpomene* and posterior probability of

classification was studied using linear discriminant analysis (LDA) as implemented in the R package MASS. The results have been adjusted accordingly.

5- About perfect mimicry

L40-41. I think that the concept of Mullerian mimicry deserves a better explanation for the unfamiliar reader since it is central to your point.

I was also surprised, given that the article is dealing with the matter of “perfect mimicry” not to see any ecological aspects on that point or paper on the evolution of mimicry. I know this is briefly mentioned in the last paragraph of discussion but it should be in introduction to assess whether selection is indeed expected to favour perfect mimicry (wouldn't coarse resemblance be enough??).

See perhaps some of those references.

Leimar, O., Tullberg, B.S. & Mallet, J. (2012) Mimicry, saltational evolution and the crossing of fitness valleys. *The Adaptive Landscape in Evolutionary Biology* (eds E.I. Svensson & R. Calsbeek), pp. 259–267. Oxford University Press, New York, NY, USA.

Ihalainen, E., Lindstrom, L., Mappes, J. & Puolakkainen, S. (2008) Can experienced birds select for Mullerian mimicry? *Behavioral Ecology*, 19, 362–368.

Ihalainen, E., Rowland, H.M., Speed, M.P., Ruxton, G.D. & Mappes, J. (2012) Prey community structure affects how predators select for Mullerian mimicry. *Proceedings of the Royal Society B-Biological Sciences*, 279, 2099–2105.

Penney, H.D., Hassall, C., Skevington, J.H., Abbott, K.R. & Sherratt, T.N. (2012) A comparative analysis of the evolution of imperfect mimicry. *Nature*, 483, 461–464.

Ruxton, G.D., Franks, D.W., Balogh, A.C.V. & Leimar, O. (2008) Evolutionary implications of the form of predator generalization for aposematic signals and mimicry in prey. *Evolution*, 62, 2913–2921

>> Thank you for these helpful references.

We improved the introduction of Müllerian mimicry as follows:

- “Here, we aim to test the possible causes of imperfect mimicry by comparing the exact color pattern mismatch between *Heliconius erato* and *Heliconius melpomene* populations and contrasting these differences with *WntA* KO phenotypes. Although there is no evidence for gene flow between *H. erato* and *H. melpomene*, which split around 12-14 Mya [22], their resemblance in wing colour patterns is remarkable. This phenotypic convergence has evolved through strong selection pressures that benefit a common warning pattern that birds have learned to associate with unpalatability [23,24]. Likely resulting from the discriminatory visual properties of birds, fine scale adjustments to the shape and size of color patterns have been observed in local mimetic butterfly communities, suggesting that the strong selection coefficients for the color patterns can force genetic variation to trace local phenotypic optima (for an overview of selection coefficients see [25]) [26,27].”

We changed the following paragraph in the introduction to introduce alternative hypothesis that could explain imperfect mimicry and that are goal is to separate these:

- “Not all study systems allow to perform artificial selection experiments. Alternative to such controlled experiments, cases of convergent evolution provide natural opportunities to investigate the selective, genetic and developmental routes to adaptation [1]. For example, mimicry and the resulting phenotypic convergent evolution between distinct butterfly species provides a comparative framework to investigate the genes underlying the evolution and diversity of a wing color pattern [12–14]. Recent studies have shown, for example, that the genes *WntA*, *cortex* and *optix* are repeatedly used to control variation in wing color patterns across Nymphalid butterflies [14–17]. Nevertheless, even in cases of Müllerian mimicry between species within the *Heliconius* genus in which both partners have used the same gene to converge on an aposematic warning signal, some degree of imperfection in resemblance may exist. However, the precise extent to which *Heliconius* mimetic butterflies need to perfectly resemble the same phenotype to maximize the fitness value is not well understood. What may underlie these differences in resemblance are (1) conflicting selection pressures and/or (2) genetic and developmental constraints. Conflicting selective pressures can include variation in the mimicry community [18], relaxed selection due to coarse discrimination by predators [19,20] and conflict between the outcomes of natural and sexual selection [21]. On the other hand, genetic and developmental constraints can result from divergence in the genetic background or in the assembled gene regulatory network that affects the functioning and phenotypic effect of these genes.”

Moreover, the results presented L206-209, that relative size of the colour area does not significantly vary between co-mimetic pairs is really interesting on the point of view of mimicry resemblance! Does this mean that when considering relative size (L207 “size” is it absolute or relative size?) variability between species is rarely more than variability between individuals. I think that this work on relative surface coloured brings another dimension to the paper that will raise interest not only from geneticist but also from ecologist. As discussed at the end of the discussion, that question whether the position or the accuracy of the patch is important or whether simply the global ratio colour/wing is not already good enough for mimicry. On the question genetic vs. selection, it means that, given the genetic constraints on some parts of the wings, other ways to enlarge the colour patch have emerged leading to convergence in surface... I wonder whether it would not be worth bringing back Fig S5 (or an improved version of it that takes into account intra-group variability into the main text?

>> We agree this is a very interesting observation. We incorporated Fig S5 into Fig 2.

#6 Figures are beautiful but the legend is hard to follow sometimes.

Fig. 1 I can't understand well the wing legends along the axis. I know from experience that such kind of variation is hard to represent... Perhaps simply trying to be more explicit by what “predicted” means?? Also since positive/negative image are a mirror of each other, do we need two wings per axis? + see my remark below on “expression of the pattern” = “presence of colour”!

>> We changed “predicted” in figure 1 to “PCA predicted presence of color”. Regarding two wings per axis, we believe it is slightly harder to interpret the variation along the axis if only one wing were given, but we do agree that the wings along the PCA axis present the inverse of each other, given the linear transformation of the data.

Fig 2 same remark “proportion” below the 1st heat band could be “proportion of individuals with colour”?

>> Changed to “Proportion of individuals with color”.

- Minor comments:

L90 “Images were obtained through the authors’ collections and collections made publicly available by Cuthill et al and Jiggins et al”

I couldn’t help but noticing that some of the samples listed in Table S2 have been collected by myself (and M. Joron) and photos taken by myself... I am slightly surprised, but of course more than happy that those data are useful for more interesting studies. Perhaps the authors needs to update this sentence or the acknowledgments? :-)

>> Our apologies. They seeped into the collection that I assumed was from Chris Jiggins, through the collaboration with Markus Moest. We checked the list, contacted everyone who has shared pictures that may not be in the earthcape repository and hope the sentence and acknowledgements are now appropriate.

L94: “allometric”. I am really puzzled by the use of this term throughout the manuscript. To me, and for all I could have read, allometry generally refers to relationship with size. Here, I don’t see any analysis of size. Do the authors mean that the landmark excluded are the one that describe inter-specific differences of shape (regardless of pattern). Am I right? Please reconsider the use of the word “allometric”, or if I am wrong, please define clearly this term for other readers; (in morphometrics it is always used for relationship with size).

>> We indeed misused this term and corrected to only refer to inter-specific shape differences.

Methods: how does the analysis takes into account the fuzziness of the border of the colour patch? Is there a threshold to draw the limit between colour and black? In my experience, *H. erato* has usually very sharp borders between the colour and the black while *H. melpomene* has a mix of red and black scales.

>> Patternize extracts, in this particular case, pixels from the images that can be classified as red (i.e. are within a predetermined red color range). In theory, the butterfly wings can also be considered a pixel raster, with scale cells only being red or black (not a mixture of pigments). However, the extraction of the fuzzy boundaries can become somewhat obscured when the resolution of the images is low, which

dilutes the fuzzy boundaries into 'mixed' pixel values. We agree that in this case some of the fuzzy transition from red to black might be missed. However, upon visual inspection, the fuzzy boundaries in *H. melpomene* can be well observed in the extracted color pattern rasters (see images below) or e.g. the *H. melpomene* postman heatmaps in Figure 2. We believe this is appropriate for the questions we are aiming to address. Although, it is interesting that the fuzzy boundary in *H. melpomene* coincides with the mismatch area in the *WntA* KO's and perhaps relates to changes in regulatory gene network involved in the difference.

Example *H. e. hydara* (French Guiana; sample ID BC0004)

Example *H. m. melpomene* (French Guiana; sample ID 10428371)

L126: "higher predicted expression of the pattern".

I think the term "expression" is misleading since it can be confounded by the region where a gene is expressed?? (cf L108 "focused on the area of the forewing in which *WntA* is not expressed" ...??). My guess here is that it means "presence of colour" ? because what the authors look at is the position and size of a colour patch on a black background right?

I think the authors should aim for a more direct and simpler vocabulary. It could be "the colour pattern displayed" for L136 or "positive values represent a higher frequency in the presence of the colour"? for L126? Same in figure legends "proportion of the wing in which the trait is expressed". Wouldn't it be simpler for the reader "proportion of the wing that is coloured"?

>> We completely agree.

We changed the figure legends and line 126 to:

- “The PCA visualizes the main variations in color pattern boundaries among samples and groups and provides predictions of color pattern changes along the PC axis. In the visualization of the predicted color pattern changes along the PC axis, with positive values presentpresenting a higher predicted expressionpresence of the MFB pattern, whereas and negative values presentpresenting the absence of the pattern.”

We changed line 136 to:

- “the colour pattern displayed”.

We changed the part at line 108 to:

- “While WntA is involved in black scale development in both the fore- and hindwing in *Heliconius* [12,16], differences in the distribution of WntA have mainly been found to correlate with the position of black color in the central part of the forewing among *Heliconius* races, here called mid-forewing band or MFB [33]. With the interest of studying variation in the MFB, we extracted and focused on the area of the forewing in which black is absent and in which WntA is thus likely not expressed. MFB patterns were extracted and aligned using the R package *patternize* [34].”

Results part b: I suggest splitting the paragraph into sub paragraphs because as it stands now it is hard to follow. Perhaps explicitly split the results on the different kinds of co-mimics, then the results on surface/absolute difference? See how to integrate results on inter-individual variability?

>> We now organized this section as follows: (1) Interspecific differences based on PCA, (2) interspecific differences based on heatmap comparisons, (3) comparison of absolute difference to relative size of MFB.

L205 “average absolute difference in the MFB pattern” -> please recall that this difference takes into account position.

>> Changed sentence to:

- “In all co-mimetic comparisons of *H. erato* and *H. melpomene*, the average absolute difference which includes the position in the MFB pattern was larger than the average difference in the size of the MFB pattern (i.e. proportion of the wing in which MFB is present; Figure 2B).”

L269 “Divergence” lack a “D”

>> Fixed

Discussion

I feel that the second part of paragraph (a) and paragraph (b) have more to do with each other than the two paragraphs joined under title (a)...? Could they be joined and this would provide space for a 1st

paragraph of the discussion more focused on the phenotypic data.

I suggest reorganizing the discussion to discuss (a) actual data on pattern, variability, consistency between races, why this leads to the conclusion of constraints for some areas, why not for others maybe (some boundaries seems to be as variable between species than within species). (b) gene regulatory network. (c) implication for mimicry evolution/selection, convergence via other roads toward same surface, etc...?

>> We thoroughly revised our discussion into the following sections, which we hope reflects your suggestion:

(a) *Patterns of advergence in MFB*

(b) *Indications of genetic and developmental constraints*

(c) *Candidates of divergence in the gene regulatory network*

L263 “pattern similarity between co-mimetic *Heliconius* butterflies has been mostly described qualitatively”

\ But see: Mérot et al 2016 that the authors quote below.

>> Apologies for neglecting this study. We deleted this statement here and improved discussion of Mérot et al 2016 as well as Rossato et al 2018.

Fig S1 : a typo on panel C, both axis are labelled PC2, while the x-axis is likely Pc1.

Suppl. Fig: please simplify vocabulary like in main figures.

>> Fixed

Fig S5: it is unclear to me how absolute differences can be percentage?

>> This figure has now been incorporated into figure 2 and the legend changed to:

- “Comparison of the difference in average size of the MFB (as proportion of the wing in which MFB is present) and absolute mismatch between *H. erato* and *H. melpomene* MFB.”

Legend of Fig S3 and S4 are reversed in the text.

>> Fixed

Thank you for this interesting manuscript,

Claire Mérot

>> Thank you, Claire. These comments were extremely helpful and thoughtful.

Referee: 3

Comments to the Author(s)

The article called "Perfect mimicry between Heliconius butterflies is constrained by genetics and development" evaluated interesting aspects from imperfect mimicry in Heliconius species be associates to genetic developmental constraints. The authors used landmarks and MFB to assess the wing pattern, and then, evaluated the relationship using WntA CRISPR/Cas9 KO phenotypes obtained by Concha et al. (2019). Finally, the authors found that genetic and developmental constraints avoid the perfect convergence to mimetic.

In my opinion, there are some points that should be carefully revised by the authors to improve the quality of this article, following below:

Major points

1) The weakest points of this article are related to the statistical analyses. Although the authors pointed that "Using quantitative measurements, we first show that all co-mimetics exhibit consistent differences in their mid-forewing colour pattern shapes" and "Comparing wing shape between males and females showed no apparent difference in sex in both *H. erato* and *H. melpomene* (Figure S1C)." I do not believe the authors chose the best statistical analysis for evaluating the mimicry convergence and sex differences. The PCA analysis is important to visualize the patterns, working as an exploratory analysis. Although, the authors should run statistical analysis appropriate to assess the difference in wing pattern evaluated in this article. Please see Rossato et al (2018), this article has a similar approach and could give some ways to conduct the statistical analyses for multivariate data to assess mimicry convergence and compare male to female wings.

>> Thank you for pointing us to the Rossato et al 2018 paper. We are embarrassed to have missed this study. The study is a very important comparison to our work and provides a most helpful guide for more appropriate statistical analysis.

For both the landmark analysis and the wing color patterns, we have now tested the effect of population, sex and species on shape variables by using multivariate analysis of variance (MANOVA) as implemented in R v3.5. For this, we used only the values of samples along significant PC axes as determined by the *permutationPA* function in the R package *jackstraw* (as advised by reviewer 2). Shape discrimination between *H. erato* and *H. melpomene* and posterior probability of classification was studied using linear discriminant analysis (LDA) as implemented in the R package MASS.

The results have been adjusted accordingly.

2) It was not clear which statistical analysis was used for evaluating the MFB differences between co-mimics included in results (lines 204-215). Please include the statistical approach in the materials section.

>> Differences in the proportion of the wing in which the MFB is present was tested between co-mimics using a two-sample t-test in R. We now also tested the absolute difference in MFB (also including the effect of position) using a one-sample t-test. This has now been added to the methods and Figure 2.

4) Although the authors said they used a quantitative approach, it seems that they based the interpretation in the qualitative description in: "Mismatch between co-mimics coincides with developmental *WntA* boundaries".

>> We compared the quantitative differences between wild-type co-mimics to quantitative analysis performed on the 10 *WntA* KO wings. For ease of visualization, we chose to extract the *WntA* KO area as present in at least 2 mutant butterflies and indeed use this as discrete developmental boundaries to assess the differences we find between *H. erato* and *H. melpomene* mimics.

5) I was expecting that the last paragraph from the introduction included hypotheses associated with the objectives. However, the authors pointed out the conclusions found in the article.

>> We agree and have improved our last paragraph to better introduce our hypothesis:

- “In the current study, we aim to explore the contribution of genetic and developmental constraints on the phenotypic mismatch between comimetic pairs of *Heliconius* butterflies. We therefore first quantitatively measured and compared differences in the mid-forewing band (MFB) pattern of 14 co-mimicking populations of the butterfly species *H. erato* and *H. melpomene* from Central and South America. We then interpret these differences in light of the recent *Heliconius WntA* CRISPR/Cas9 KO mutants [12]. We were specifically interested to test how species-specific divergence might impact the developmental function of *WntA* and limit adaptive convergence. Our results suggest the possible existence of constraints, imposed by genetic and developmental differences, for natural selection to achieve perfect mimicry. By taking advantage of the most recent methodological and technological advances provided by functional experiments and computational quantitative measures, our study offers an alternative approach to artificial selection experiments to test the relative constraint of genetics and development to adaptation.”

6) At the results section, the authors discussed the results (for example, lines 212-213 and 236-239). I think it is better to keep this to the discussion section.

>> We moved these statements to the discussion.

7) I suggest the authors review the discussion after running the statistical analysis. I believe the interpretation of some of the results could change.

>> We ran the appropriate statistical analysis as suggested. We indeed found additional results that indicated sex differences.

The following results were added:

- “Comparing wing shape between males and females using the subset landmark set showed significant differences in sex in both *H. erato* and *H. melpomene* mostly along the first PC axis ($F = 14.0$, $p < 0.001$; Table S3; Figure S1C). However, sex had generally a low posterior probability of classification of 67.6 % and 63.2 % for males and females, respectively, indicating large overlap in the phenotypes of the sex classes.”
- “As for wing shape, significant differences between male and female MFB patterns were observed ($F = 3.17$, $p = 0.001$). However, sex differences had a low probability of posterior classification (55.8 % and 58.3 % for males and females, respectively) and were only significant along PC axes that explain small amounts of variation among samples (Table S3).”

Despite these significant results, removing females from our dataset did not change our results (Figure S4).

8) Please, give a background about genetic architecture associated with phenotype, since it was not completely clear in the present article.

>> In the introduction, we opted to keep the description of the genetic architecture of coloration in *Heliconius* concise and mostly focused on the *WntA* gene (which is described in a plentitude of review articles), as follows:

- “For example, mimicry and the resulting phenotypic convergent evolution between distinct butterfly species provides a comparative framework to investigate the genes underlying the evolution and diversity of a wing color pattern [12–14]. Recent studies have shown, for example, that the genes *WntA*, *cortex* and *optix* are repeatedly used to control variation in wing color patterns across Nymphalid butterflies [14–17].”
- “*WntA* is a member of the Wnt family of signaling ligands and a key molecular tool for butterfly wing color pattern development. While the gene coding sequence is highly conserved across lepidoptera, its *cis*-regulatory diversity underlies wing pattern shape variation between and within butterflies species [28,29].”

In the discussion we elaborate on the complexity of this genetic architecture in the section ‘Candidates of divergence in the gene regulatory network’.

9) The results found using the distinct data sets are a little bit disconnected. Please, make sure they are linked to the same issue. Then, connect the interpretation between wing aspects measured using landmark and MFB to the data set using CRISPR phenotype.

>> We have improved the connection between the different datasets used by significantly rewriting the manuscript. We provide a better outline of our hypothesis and a better description of how we compare the results found in the interspecies MFB comparisons and the *WntA* KO’s.

10) Please include in the discussion others selective forces that could work in distinct directions against the convergent evolution, as proposed by Estrada and Jiggins (2008), and, discuss the results found by (Rossato et al 2018) according to the imperfect convergence in *Heliconius* co-mimetic and sexual selection.

>> We added the following to the introduction:

- “However, the precise extent to which *Heliconius* mimetic butterflies need to perfectly resemble the same phenotype to maximize the fitness value is not well understood. What may underlie these differences in resemblance are (1) conflicting selection pressures and/or (2) genetic and developmental constraints. Conflicting selective pressures can include variation in the mimicry community [18], relaxed selection due to coarse discrimination by predators [19,20] and conflict between the outcomes of natural and sexual selection [21]. On the other hand, genetic and developmental constraints can result from divergence in the genetic background or in the assembled gene regulatory network that affects the functioning and phenotypic effect of these genes.”

In reference to Rossato et al 2018 and Merot et al 2016 we added the following to the discussion:

- “In our comparisons we observed that even though the position of forewing pattern elements may not be perfectly identical between co-mimics, the relative amount of black versus red or yellow is generally more similar than the absolute difference. This improved match of the size of the MFB seems to result from compensatory changes in the MFB pattern and are in line with phenotypic evolution being driven by predation pressure. A similar ‘advergence’ process has previously been demonstrated for the mimicry ring including *H. timareta thelxinoe*, *H. e. favorinus* and *H. m. amaryllis* in Peru [26] and the mimicry ring including *H. e. phyllis*, *H. besckei*, *H. m. burchelli* and *H. m. nanna* in Brazil [27]. They thus indicate fine scale pattern adaptation in non-homologous regions of the wing to obtain a better match in the shape and size of the pattern even though they have a shifted position in the wings.”

Minor points

Line 146 and (2) and - exclude one "and"

>> Fixed

Line 269 - Include D to Divergence

>> Fixed

References

Estrada, C., and Jiggins, C. D. 2008. Interspecific sexual attraction because of convergence in warning coloration: Is there a conflict between natural and sexual selection in mimetic species? *Journal of Evolutionary Biology*, 28, 749–760.

Rossato, D. O., D. Boligon, R. Fornel, M. R. Kronforst, G. L. Gonçalves, and G. R. P. Moreira. 2018. Subtle variation in size and shape of the whole forewing and the red band among co-mimics revealed by geometric morphometric analysis in *Heliconius* butterflies. *Ecology and Evolution*.

Appendix B

Referee: 2

Comments to the Author(s).

This is a revised version of an interesting manuscript that I have reviewed before, and I am still very enthusiastic about its relevance and interest.

The authors have nicely and deeply improved the manuscript, which stands stronger and much easier to read. They have addressed all of my concerns appropriately and, to the extent of my knowledge, this is also true for other's reviewers' remarks.

I made below a couple of suggestions to improve clarity but those are really minor.

Claire Mérot

>> Thank you very much for going through our manuscript with great detail a second time.

- Minor points

Abstract: I feel that the abstract is long and vague about the context (particularly L14-17) without highlighting the exciting results (same area repeatedly divergent between pairs of mimics – overlap with KO boundaries)... Perhaps some re-writing to have more impact?

>> We have shortened rewritten the abstract following your advice as follows:

“Müllerian mimicry strongly exemplifies the power of natural selection. However, the exact measure of such adaptive phenotypic convergence and the possible causes of its imperfection often remain unidentified. Here, we first quantify wing color pattern differences in the forewing region of 14 co-mimetic races of the butterfly species *Heliconius erato* and *Heliconius melpomene* and measure the extent to which mimicking races are not perfectly identical. Next, using recent CRISPR/Cas9 KO experiments of the gene *WntA*, which has been mapped to color pattern diversity in these butterflies, we explore the exact areas of the wings in which *WntA* affects color pattern formation differently in *H. erato* and *H. melpomene*. We find that, while the relative size of the forewing pattern is generally nearly identical between co-mimics, the CRISPR/Cas9 KO results highlight divergent boundaries in the wing that prevent the co-mimics from achieving perfect mimicry. We suggest that this mismatch may be explained by divergence in the gene regulatory network that defines wing color patterning in both species, thus constraining morphological evolution even between closely related species.”

L68: It is really strange to have “here , we aim..” and then again “L93 “in the current study, we aim”. I'd removed L68-70 to rather jump into presenting the system and keep L93 the exact scope of the study.

>> We removed L68-70.

L85: “a textbook example of Mullerian mimicry” -> either already above or comes a bit late? Perhaps this whole part could be L68 to introduce the system and the question.

>> We moved this statement to the previous paragraph as suggested.

L114 “describe”-> described

>> Fixed

L144: “5 mutants”? how many per species?

>> Changed to “...Five mutant butterflies for each of the Panamanian geographic races...”

L168: what about other races than postman?

>> Differences between non-postman *H. erato* and *H. melpomene* races are also highly significant ($F = 64.8$, $p < 0.001$), with a posterior probability of classification of 92.5 % and 97.6 % for *H. erato* and *H. melpomene*, respectively. However, for conciseness we opt to not report this and focus on the differences in the postman phenotypes, which builds the argument for later analysis and comparison to the mutant phenotypes.

L173: Why “as for wing shape”? -> not mentioned before

>> We removed this statement, which is discussed in Supplementary materials 1.

L175: 55% or 58% of classification is indeed low. To reject more robustly any strong effect of sex, it would be good to show that 55% of classification is in fact similar to classification between random groups (this could be done just by resampling two groups of similar sample size as male and female but mixing the sex label, and ask what % of classification is obtained. I suspect this would be a range of 40-60% meaning that in fact the sex effect on pattern is unlikely to affect the results)

>> A random assignment of groups indeed results in an expected classification of 50% for both males and females with a standard deviation of 7 for males and 12 for females, due to the higher number of males in the dataset. We tested this with 100 permutations. We modified the text as follows and also split the sex significance testing for *H. erato* and *H. melpomene*:

“Significant differences between male and female MFB patterns were observed in both *H. erato* ($F = 3.11$, $p = 0.002$) and *H. melpomene* ($F = 3.17$, $p = 0.001$). However, sex differences had a low probability of posterior classification (55.8 % and 58.3 % for male and female *H. erato* and 78.0 % and 62.5 % for male and female *H. melpomene*, which is close to classification between random groups) and were only significant along PC axes that explain small amounts of variation among samples (Table S4).”

L179-181: cool result! And it is now nicely visible with the orange triangle

>> Thanks!

L202: I wonder whether this sentence about the control of consistent results with a diff set of landmarks

cannot go into supplementary or methods to streamline the results?

>> Agreed. We moved this sentence to Supplementary materials 1.

L238 “One of the most striking results that emerged from the KO phenotypes is the divergent developmental architecture underlying butterfly wing color pattern convergence”

-> I don't get that at all. The KO confirm that the same gene (*WntA*) controls the same trait in both species (when look broadly). That looks like a similar (conserved? Or convergent?) architecture to me ... Should make it easy to be mimics... But then the exact domains affected by this gene does vary -> hence the question of whether divergence at *WntA* or related network or background effect explain imperfect mimicry. Perhaps be more precise about the scale at which there is convergence and divergence?

>> The point we are trying to make here is that despite *WntA* being mapped to forewing band variation in both *H. erato* and *H. melpomene*, there are other factors that have diverged and changed the role of *WntA* between these species. This means that, for example, other factors must control black scale development in the proximal region of the wing in *H. m. rosina* compared to *H. e. demophoon*. To be more clear, we modified the statement as follows:

“One of the most striking results that emerged from the KO phenotypes is an apparent divergent developmental architecture underlying butterfly wing color pattern convergence, despite *WntA* being involved in forewing band variation in both species.”

Fig 2: nicely improved!! However, it would be appreciated to have both the limits of the erato KO and the Melpomene KO on the last column. This difference is at the core of the study and the main conclusions.

>> Agreed. We added the *H. m. rosina* KO outline and changed the color scales for better contrast.

Referee: 1

Comments to the Author(s).

Van Belleghem et al. have revised this manuscript and it is now considerably improved, much clearer and more accessible. They also answered all my questions. The authors could maybe slightly improve and extend their explanation of their argument that the overlap of KO areas and wild-type differences between species suggests the presence of different developmental networks. It's a key point in this MS and sufficiently explained but a slightly more elaborate explanation may be helpful for a readers less familiar with this topic.

>> We tried to improve this message by adding the following sentence to the abstract:

“We find that, while the relative size of the forewing pattern is generally nearly identical between co-mimics, the CRISPR/Cas9 KO results highlight divergent boundaries in the wing that prevent the co-mimics from achieving perfect mimicry.”

We modified the last paragraph of the introduction as follows:

“We were specifically interested to test how species-specific divergence might impact the developmental function of *WntA* and limit adaptive convergence. More precisely, we argue that overlap of wild-type differences with differences in pattern boundaries as defined by *WntA* KOs in *H. erato* and *H. melpomene* would suggest the possible existence of genetic and developmental constraints for natural selection to achieve perfect mimicry.”

We added the following statement to the results:

“We focused on the Panamanian co-mimics *H. e. demophoon* and *H. m. rosina* for which the largest *WntA* KO dataset is available and compared the *WntA* boundaries defined by these mutants with wild-type variation in the postman phenotypes.”

I also have a few additional minor comments:

L22: „define“ should be „defines“ or „network“ should be „networks“

>> Fixed

L59: better „genes“?

>> Changed to “genes”

L64: „relaxed selection due to coarse discrimination by predators“ – important point but I am not sure relaxed selection falls into the category „conflicting selection pressures“?

>> Agreed. We changed this to:

“What may underlie these differences in resemblance are (1) conflicting or relaxed selection pressures and/or (2) genetic and developmental constraints. Conflicting selective pressures can include variation in the mimicry community [18] and conflict between the outcomes of natural and sexual selection [21]. Relaxed selection pressures may result from coarse discrimination by predators [19,20].”

L90: I think both divergence in cis-elements as well as trans-elements affect epistatic interaction (e.g. trans-elements bind to cis-elements) but the line seems to suggest that epistatic interactions are always trans?

>> Agreed. We removed cis and trans from this sentence.

L166: states „between each of the co-mimicking pairs“ but the subsequently presented statistics and results are for „between the species“, I think

>> We changed this sentence as follows:

“However, significant differences in clustering can be observed in the PCA between the two species, with posterior probability of classification 88.6 % and 92.2 % for *H. erato* and *H. melpomene*, respectively ($F = 70.8$, $p < 0.001$; Table S4; Figure 1C).”

L297: e.

>> Fixed

Referee: 3

Comments to the Author(s).

The article intituled "Perfect mimicry between Heliconius butterflies is constrained by genetics and development" assessed the convergence in wing pattern between co-mimics using innovative and sophisticate tools. I am convinced that this work will contribute to many areas of knowledge, mainly to ecological and evolutionary studies. However, there are some aspects that I believe could be improved. The main point is related to the connection between the parts of the article. I don't know if I missed, unfortunately, but the connection between the first part (Color pattern analysis) and the second part (WntA CRISPR KO analysis) of the article is not completely clear for me. I understood both separately, but I can not see how the WntA CRISPR KO phenotype help to explain the imperfection in wing colour pattern in 13 mimicking races. Is there some selective pressure selecting the pattern in the same way between all co-mimetics? Additionally, I did not understand how the inclusion of the WntA CRISPR KO phenotype of *Heliconius erato demophoon* in Figure 2 contribute to understanding the pattern of the co-mimetic, and why the authors choose the phenotype of *Heliconius erato demophoon* and no of *Heliconius melpomene rosina* to represent the mutant pattern.

>> With the suggestions of reviewer 1 and 2 we now added several sentences to the abstract, intro and results to better connect the observation of wild-type differences and KO boundaries. We believe this should help to connect the parts of our manuscript.

The races other than the postman races are indeed not directly comparable to the mutant postman phenotypes. We do however use variation and species differences of these co-mimicking populations to demonstrate that differences can be observed between all populations and that developmental constraints may thus affect the entire adaptive radiation. Within the species, the vast diversity in MFB that falls outside of the WntA KO boundaries also suggests that additional loci outside of *WntA* must have diverged.

We also now added the *H. m. rosina* mutant pattern to figure 2 and added the following to the caption:

“As a positional reference of the MFB pattern variation, the column on the right overlays the *H. e. demophoon* (green outline) and *H. m. rosina* (yellow outline) *WntA* CRISPR KO phenotype as found in at least 50 % of the KO samples with the differences between co-mimics.”

- I suggest a change in the expression "the size of the MFB" for "the value of the MFB", avoiding the confusion with the allometric relationship between size and shape.

>> In our manuscript, we use "relative size of the MFB" as the proportion of the wing in which the MFB is present. When we describe pattern differences between co-mimicking populations we use "absolute MFB differences" which include both size and position, but are controlled for changes in wing size. We now made sure to use either "relative size" or "proportion of the wing" when we refer to MFB size.

- In lines 165 -167, if I have understood correctly, the posterior probability of classification was assessed in *H. erato* and *H. melpomene*; however, I believe the posterior probability could be assessed within population (co-mimetics). Make more sense according to the objectives of the article.

>> This section demonstrates that there are species specific differences between the co-mimicking *H. erato* and *H. melpomene* populations, which is most relevant to our study. Posterior classifications are thus for classifying co-mimicking races as either *H. erato* or *H. melpomene*. Within species posterior classifications are also given in Table S4, but are not discussed in the text for conciseness. We fixed the sentence for clarity as follows:

"However, significant differences in clustering can be observed in the PCA between the two species, with posterior probability of classification 88.6 % and 92.2 % for *H. erato* and *H. melpomene*, respectively."

- Please, include degrees of freedom in results from statistic F.

>> We added the degrees of freedom to the F statistic, also in Table S3 and S4.

- In lines 174-175, the authors did not find a high posterior probability of classification in sexes; however, as many other works suggest there is a high difference between male and female, I would like to know if the authors assessed the difference between male and female within species.

>> We note that sex differences are indeed significant and observable. We argue however, that these sex differences do not correlate with between species differences or the differences observed in the KO boundaries and thus not affects the interpretation of our results. We modified the results statement as follows:

"However, sex differences had a low probability of posterior classification (55.8 % and 58.3 % for male and female *H. erato* and 78.0 % and 62.5 % for male and female *H. melpomene*, which is close to classification between random groups) and were only significant along PC axes that explain small amounts of variation among samples (Table S4)."